# An early warning signal for grassland degradation on the Qinghai-Tibetan Plateau

Qiuan Zhu [1,11] ✉, Huai Chen [2,11], Changhui Peng [3,4], Jinxun Liu [5], Shilong Piao [6], Jin-Sheng He [7], Shiping Wang [6], Xinquan Zhao[8], Jiang Zhang[1], Xiuqin Fang[1], Jiaxin Jin[1], Qi-En Yang [8], Liliang Ren[9] & Yanfen Wang [10] ✉

Intense grazing may lead to grassland degradation on the Qinghai-Tibetan Plateau, but it is difficult to predict where this will occur and to quantify it. Based on a process-based ecosystem model, we define a productivity-based stocking rate threshold that induces extreme grassland degradation to assess whether and where the current grazing activity in the region is sustainable. We find that the current stocking rate is below the threshold in ~80% of grassland areas, but in 55% of these grasslands the stocking rate exceeds half the threshold. According to our model projections, positive effects of climate change including elevated $CO_2$ can partly offset negative effects of grazing across nearly 70% of grasslands on the Plateau, but only in areas below the stocking rate threshold. Our analysis suggests that stocking rate that does not exceed 60% (within 50% to 70%) of the threshold may balance human demands with grassland protection in the face of climate change.

The Qinghai–Tibetan Plateau (QTP), colloquially referred to as the "rooftop" of the world, lies at a mean elevation higher than 4000 m, and covers around 2.5 million km², corresponding to a quarter of China's land area[1]. The QTP is also the "Asian Water Tower", providing water to 1.9 billion people, and extensive ecosystem benefits, including climate regulation, soil conservation and cultural services[2,3].

About 70% of the QTP is covered by different types of alpine grasslands, including desert steppes, alpine steppes and alpine meadows (Fig. 1), making it the highest and largest alpine grassland in the world[4,5]. Such alpine grasslands provide forage for 12 million yak and 30 million sheep and goats, and their responses and resilience to climate change and human activities strongly affect the livelihood of about 5 million pastoralists and agropastoralists on the QTP[6,7].

QTP grasslands, one of the world's harshest grazing environments, are very fragile and extremely sensitive to climatic conditions[1,8]. They face increasing impacts from climatic change, and the climate has been warming faster than in many other regions in the world[7,9]. The ubiquitous warming and overall slight wetting are dominant characteristics of climate change of the QTP that have had positive and negative effects on the alpine grasslands which subsequently have been under great changes in the past decades. Earlier in history, animal farmers tended to maintain relatively small herds just large enough to support their nomadic lifestyle, with some extra animals to buffer the effects of harsh weather. The grazing of these small herds was sustainable because it did not lead to irreversible grassland degradation, defined as a decline in the quality of plants and soils, as well as changes in ecosystem composition, structure, and function[10,11]. More recently,

[1]College of Geography and Remote Sensing, Hohai University, Nanjing 210098, China. [2]Chengdu Institute of Biology, Chinese Academy of Science, Chengdu 610041, China. [3]Department of Biology Science, Institute of Environmrnt Sciences, University of Quebec at Montreal, Montreal H3C 3P8 QC, Canada. [4]School of Geographic Sciences, Hunan Normal University, Changsha 410081, China. [5]U.S. Geological Survey, Western Geographic Science Center, Moffett Field, CA 94035, USA. [6]State Key Laboratory of Tibetan Plateau Earth System, Resources and Environment (TPESRE), Institute of Tibetan Plateau Research, Chinese Academy of Sciences, Beijing 100101, China. [7]Institute of Ecology, College of Urban and Environmental Sciences, Peking University, Beijing 100871, China. [8]Key Laboratory of Adaptation and Evolution of Plateau Biota, Northwest Institute of Plateau Biology, Chinese Academy of Sciences, Xining, Qinghai 810001, China. [9]The National Key Laboratory of Water Disaster Prevention, Hohai University, Nanjing 210098, China. [10]University of Chinese Academy of Sciences (UCAS), Beijing 100101, China. [11]These authors contributed equally: Qiuan Zhu, Huai Chen. ✉e-mail: zhuq@hhu.edu.cn; yfwang@ucas.ac.cn

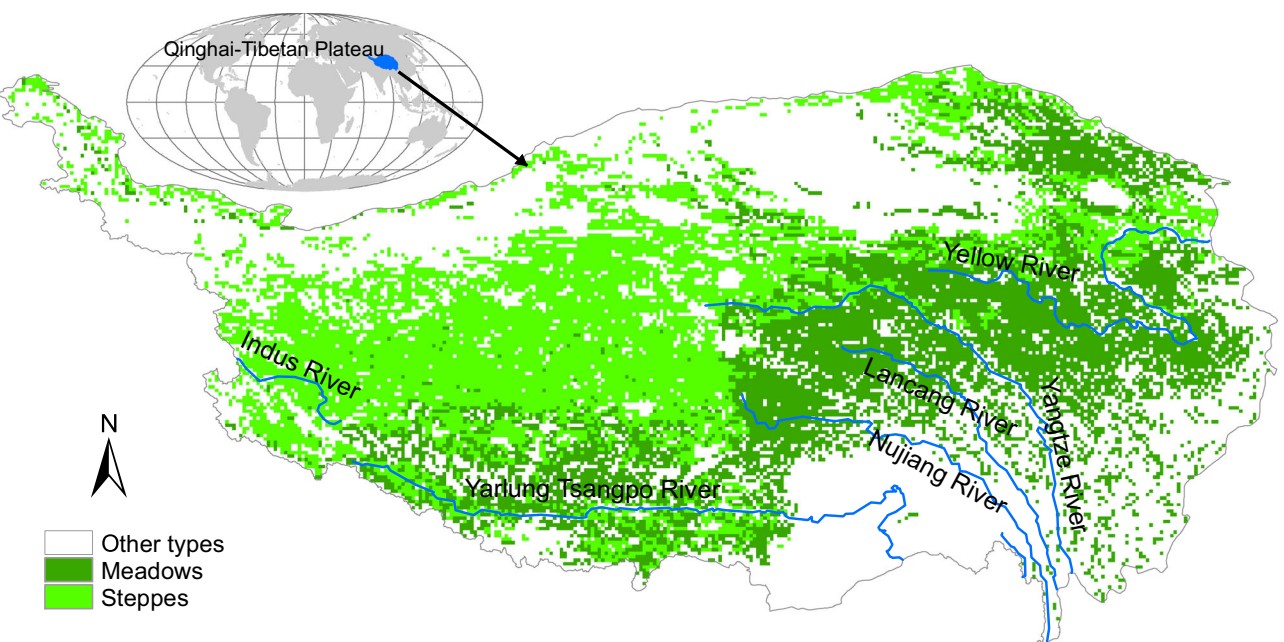

**Fig. 1 | Geographical distribution of grasslands on the Qinghai-Tibetan Plateau, based on a vegetation map of China (1:1000000).** Data source: Editorial Committee of Vegetation Map China (2007).

however, along with climate change, human activities on the QTP have intensified as the population has increased to 12 million, leading to increasing construction of roads and railways, land reclamation, and overgrazing. In particular, the massive increase in livestock grazing on the QTP appears to be a major driver of grassland degradation that reduces the quality of plants and soils[11]. Altogether, some studies showed that human activities have mainly contributed to the current degradation of the alpine grasslands on QTP[12,13].

Even slight changes in climate and human factors can substantially affect the structure and functioning of alpine grasslands on the QTP[14–16], leading to sudden, potentially irreversible changes in ecology and functioning[17,18]. With growing concern on the complex interactions between human activities and the natural ecosystem processes, how to identify the occurrence time of possible abrupt and irreversible state shifts in the alpine grassland ecosystem is essential for sustainable development and is the key for effective management in grassland ecosystem degradation mitigation[15,18].

Given the pertinence of grazing to QTP grasslands[8,19], we propose to develop a "stocking rate (expressed as the number of sheep unit per hectare of grassland per year in this study) threshold" which would induce an extreme grassland degradation to provide an early implication for grazing activities management. Since grazing in different regions varies according to local climate factors (e.g. precipitation and temperature)[2], an effective management strategy for reasonable grazing patterns and grazing period arrangement based on a predicted stocking rate threshold that could induce grassland degradation may support balancing the grazing activities and carrying capacity of the grassland on QTP[20,21]. Such a threshold may also help to guide grazing management and sustainable development of grasslands on QTP in the future.

The central Chinese government has launched several initiatives to mitigate degradation of grasslands around the country, especially on the QTP[3,20–23]. The "Grain to Green Program", initiated in 1999, began restoring grassland from cropland, while the "Retire Livestock and Restore Pastures", initiated in 2003, and the "Return Grazing to Grassland Project", initiated in 2004, began replacing grazing areas with uncultivated pastures. While these programs have achieved some success, it remains unclear how to ensure grazing patterns on QTP

grasslands that are ecologically sustainable in the long term and can support the livelihood of local populations[8]. Instead of local scale, in depth understanding of a stocking rate threshold that could induce grassland degradation at regional scale across the QTP could help to propose an early warning signal for grassland degradation to inform the management decisions related to the above-mentioned ecological projects.

Numerous studies have investigated the impact of grazing on QTP grasslands on a local scale, for example in terms of vegetation cover[24], species richness and diversity[25], above-ground primary production[25], soil organic carbon[26], soil respiration[27] and soil nutrient dynamics[28]. However, whether these local measurements can be reliably extrapolated to regional scales on the QTP grasslands is unclear, making it difficult to define an appropriate stocking threshold that induces grassland degradation base on statistical hypothesis framework or field observation work[29]. As a result, so-called "process-based ecosystem modeling", which integrates information about biogeophysics, biogeochemistry, plant phenology, vegetation dynamics, as well as cycling of carbon, water and energy on the land surface, has shown promise for simulating interactions among vegetation, climate, and human activities[30,31], making it well-suited to detect productivity changes in specific ecosystems[32]. Conducting long-term simulations at grid scale across the whole QTP using a process-based ecosystem model that consider the interaction of different processes among atmosphere, vegetation and soil could provide more insights into the spatial heterogeneity of ecosystem changes and could be extended into the past or the future to take climate change into account.

In this work, we modified a grazing framework for grassland ecosystems and integrated it into a process-based model to explore a stocking rate threshold inducing grassland degradation on the QTP under current conditions and future scenarios with climate change. We adopted net primary productivity (NPP) as an indicator of grassland degradation, which has proven to be a good estimator of ecosystem functioning[33] and land degradation[34]. We assumed that grassland productivity would decrease with increasing stocking rate, and we defined the stocking rate threshold to be the stocking rate at which NPP fell to 1% of the pregrazing level that was considered as an extreme degradation status for grassland (see Methods). The objectives of this

study were to detect a potential stocking rate threshold that could risk grassland extreme degradation on the QTP, apply the stocking rate threshold to identify grassland areas under threat of degradation, and predict when such degradation might occur under current conditions and future conditions of grazing, climate change and elevated $CO_2$. We show that the current stocking rate is below the threshold in ~80% of grassland areas on the Plateau. We further suggest that although climate change including elevated $CO_2$ can partly offset negative effects of grazing across nearly 70% of grasslands on the Plateau, stocking rate that does not exceed 60% of the threshold may balance human demands with grassland protection in the face of climate change.

## Results

### Spatial pattern of stocking rate on QTP grasslands
The multiyear average stocking rate on the QTP generally increased from northwest to southeast during the period from 1980 to 2017 (Fig. 2a). Greatest stocking rate (>10 sheep units (SU) ha$^{-1}$ year$^{-1}$) was found mainly in the east and in the south. Intermediate stocking rate was found in the southeast (5-9 SU ha$^{-1}$ year$^{-1}$). Analysis by decade showed that stocking rate increased continuously in the east (~0.1 SU ha$^{-1}$ year$^{-1}$ year$^{-1}$), especially between 1980 and 1989 (Fig. 2b–e). From 1990 to 1999, stocking rate decreased across most of the Plateau, but it has been increasing in the central and eastern regions since 2000.

### Patterns of stocking rate threshold on QTP grasslands
Stocking rate thresholds were mapped across the QTP to examine the spatial pattern of intensity of grazing activities that possibly inducing extreme grassland degradation (Fig. 3a). Stocking rate thresholds were highest in the east (greater than 9 SU ha$^{-1}$ year$^{-1}$), while they were lowest in the northwest (mostly less than 1.5 SU ha$^{-1}$ year$^{-1}$) (Fig. 3a). In general, thresholds are lower in the northwestern part and higher in the southeastern part of QTP. The stocking rate threshold fell between 1 and 4 SU ha$^{-1}$ year$^{-1}$ over most of the Plateau (Fig. 3b).

Next, we assessed where current stocking rate lay below or above the local threshold. We found that approximately 80% of the QTP grassland area had a stocking rate below the threshold, particularly in central regions (Fig. 3c). Nevertheless, stocking rates in most of those areas (~55%) were greater than half of the threshold values (Fig. 3d). Areas where actual stocking rate exceeded the threshold, usually within a factor of two (Fig. 3d) were within the same general area as those showing high stocking rate (Fig. 2a).

### Time until degradation of QTP grasslands at different stocking rates
Given that stocking rate on most of the QTP appears to lie under stocking rate threshold, we wanted to know how long it would take until grassland became degraded in different areas if we assumed that stocking rate remained at the threshold level, or even higher than the threshold (10% or 30% higher). If stocking rate remained at the threshold, model results projected that the grasslands in the northwest would become degraded within 20 years, compared to more than 80 years for grasslands in the south and southeast (Fig. 4a). Most grasslands on the QTP would become degraded within 40–80 years (Fig. 4d). In general, model results predicted that longer time for grasslands to become degraded as one moved from the northwest to the southeast under the grazing activities at intensity of threshold (Fig. 4a). If stocking rate remained at 10% or 30% above the threshold, we predicted that most grasslands would become degraded within, respectively, 20–50 years or 10–20 years (Fig. 4b, c, d). Generally, grassland degradation accelerated by approximately 25 years or 40 years for these two conditions respectively while comparing the degradation time under the condition exactly with the threshold (Fig. 4d).

The areas where current stocking rate exceeded the threshold were mainly located in the northwest, southern and northeast parts of QTP (Fig. 4e, f). Modeling of these areas indicated that 96.2% of the overgrazed grassland would be degraded within 40 years (Fig. 4f) or 83.0% within 20 years if stocking rate remained at actual levels (Fig. 4e). Since we applied the grazing activities in the simulation from 1980 onwards, it indicated that the grasslands in these areas would be degraded by now. If we reduced stocking rate in these areas to the threshold level, the time until degradation would be more than 60 years in the southern and northeast parts, but still fewer than 10 years in the northwest part (Fig. 4f), which implies that the southern and northeast grasslands of the QTP can currently be considered undegraded, while the northwest grasslands can still be considered degraded.

### Ratios of actual stocking rate to the threshold on QTP grasslands
Simulations suggested that climate change and elevated atmospheric $CO_2$ concentration would partly offset the negative effects of grazing activities on 68.3% of QTP grasslands (Fig. 5a, b and Supplementary Note 1), nearly all of which (98.5%) was subject to normal grazing and which accounted for 83.6% of the total grassland area subject to normal grazing (Fig. 5b). Across these "offset" areas, 69.2% showed a ratio of stocking rate to the threshold below 0.6, while 83.0% showed a ratio below 0.7 (Fig. 5c). The beneficial offsets were observed in only 5.3% of overgrazed areas (Fig. 5b), 76.9% of which featured ratios of stocking rate to the threshold below 3.0 (Fig. 5d). Among the remaining 31.7% of QTP grassland areas where climate change and elevated $CO_2$ were not predicted to offset negative effects of grazing (Fig. 5b), 58.5% were overgrazed (Fig. 5b), with ratios of stocking rate to threshold from 2.0 to 6.0 (Fig. 5e), and 41.5% were subject to normal grazing (Fig. 5b), of which 71.3% showed a ratio of stocking rate to threshold larger than 0.6 and 88.4% showed a ratio of stocking rate to threshold larger than 0.5 (Fig. 5f). Although 41.5% of "non-offset areas" were under normal grazing, the positive effects of climate change and elevated $CO_2$ could not compensate the negative effects of grazing on grassland productivity.

Our model suggested that setting the ratio of stocking rate to threshold between 0.5 and 0.7 would preserve 70–80% of the areas that were currently subject to normal grazing and were predicted to experience positive offsets from climate change and $CO_2$ (Fig. 5c, g). This range from 0.5 to 0.7 showed the greatest potential for maximizing the areas subject to normal grazing or overgrazing (both currently in "non-offset areas") that could benefit from positive offsets due to climate change or $CO_2$ (Fig. 5b, e, f). When we reduced the ratio of stocking rate to threshold to 0.6 in areas where the ratio was >0.6 in an additional simulation, we found that the positive effects of climate change and elevated $CO_2$ could offset the negative effects of grazing on over 89.1% of QTP grassland areas (Fig. 5h), by "converting" 70% of currently "non-offset areas" into "offset areas" (Fig. 5b, h). The "converted" areas come from 78% of overgrazing area and 58% of normal grazing area (Fig. 5b, h).

### Future patterns of stocking rate threshold on QTP grasslands
In a further effort to understand the possible changes of stocking rate threshold of QTP grasslands in light of future climate change, we predicted stocking rate thresholds for current condition (Fig. 6a) or under different climate scenarios as defined by representative concentration pathways (RCPs) 2.6, 4.5 and 8.5 (Fig. 6b–d). In all three scenarios, stocking rate thresholds generally increased from northwest to southeast, and they increased with radiation forcing (RCP 8.5 > 4.5 > 2.6), particularly in the south and southeast. Compared to the current climate status, thresholds under all the RCP scenarios were larger, and thresholds > 4.0 SU ha$^{-1}$ year$^{-1}$ covered a larger surface area (Fig. 6e). Grassland surface area with a threshold > 7.0 SU ha$^{-1}$ year$^{-1}$ was substantially greater under the scenario RCP 8.5 than under any other scenarios (Fig. 6e).

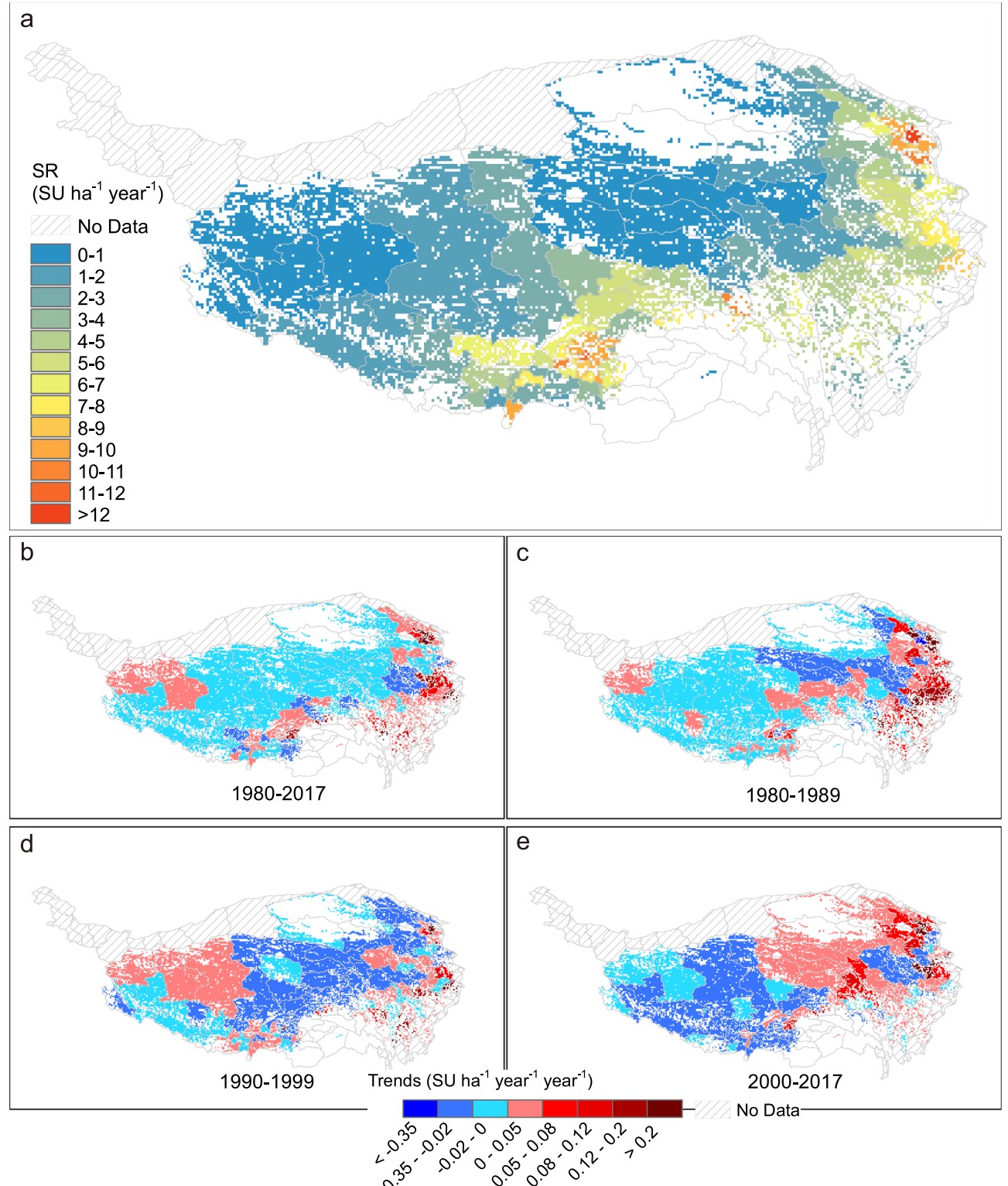

**Fig. 2 | Stocking rate across grasslands on the Qinghai-Tibetan Plateau (QTP). a** Multiyear average stocking rate at the county scale. **b–e** Stocking rate trends during different periods between 1980 and 2017. Maps were masked with an initial grassland distribution as showed in Fig. 1. SR stocking rate, SU sheep unit.

## Discussion

Overgrazing is one of the greatest drivers of grassland degradation, among many other contributors, including climate change, harsh environmental conditions, pastureland privatization and herd sedentarization[8,35]. Since grazing activities depending on productivity of local vegetation, geographical mapping to identify areas of

overgrazing may support management decision making[3]. Here we define a stocking rate threshold based on criterion of NPP that can describe the vulnerability of grassland to overgrazing and predict "time until degradation", which may help guide efforts to protect and restore grasslands. The stocking rate threshold defined in this study differs from the concept of "carrying capacity" that can "sustainably

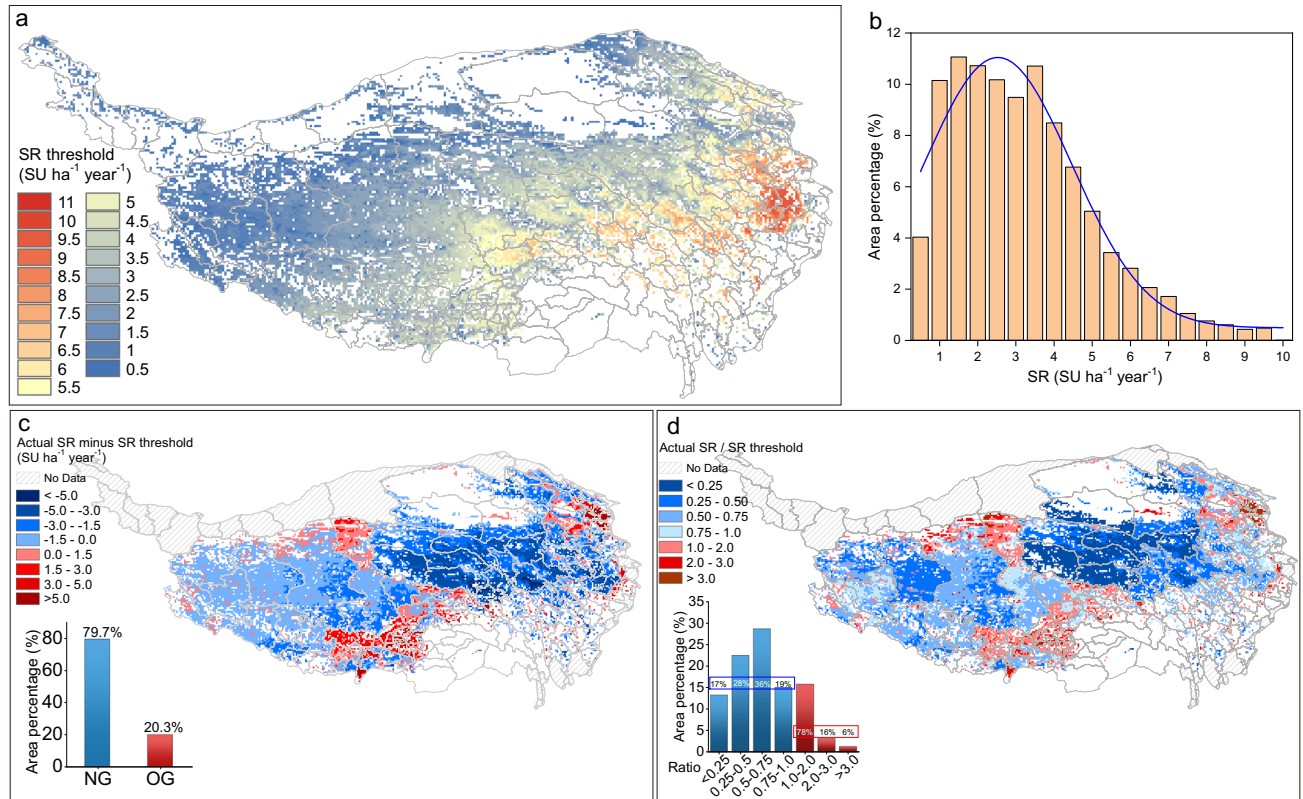

**Fig. 3 | Spatial pattern of stocking rate thresholds on the Qinghai-Tibetan Plateau (QTP) and difference between thresholds and actual stocking rate.** Maps were masked with the initial grassland distribution shown in Fig. 1. **a** Stocking rate thresholds. **b** Distribution of threshold values in terms of amount of surface area. Discrepancy between stocking rate thresholds and actual stocking rate, expressed as (**c**) absolute differences or (**d**) ratios, where the bar chart indicated the area percentage of overgrazing (red) and normal grazing (blue) in **c** and the area percentage of different ratio range in **d** (blue and red represented ratio <1 and ≥1 separately). SR stocking rate, SU sheep unit, OG overgrazing, NG normal grazing.

support"[36]. The stocking rate threshold may exceed the carrying capacity because stocking rate at the threshold level is predicted to lead to an extreme degradation which is "unsustainable". Although many previous reports showed the entire degraded percentages of grassland on the QTP, the mentioned percentages were simply taken from earlier work that did not undergo peer review (Supplementary Note 2). In fact, grassland degradation could be divided into different catalogs, including lightly degraded, moderate degraded and heavily degraded. In this study, we focus on heavily degraded scenarios, which would cause ecosystem collapse. Our simulations suggest that overgrazing and degradation affect around 20% of QTP grasslands, based on ratios of stocking rate to the threshold. This figure is comparable to the multi-year mean grassland overgrazing rates of 16% for Qinghai and 24% for Tibet between 2010 and 2017, though several studies have suggested degradation rates of 21–40% across the entire QTP from the 1980s to 2000s (Supplementary Note 2). Our results figured the most serious overgrazing status based on heavily degraded scenarios. In our analyses, estimated stocking rate threshold decreased from southeast to northwest across the QTP grasslands, reflecting the distribution of alpine meadows (in the southeast) and alpine steppes (in the northwest), partially because meadow areas had more favorable precipitation, temperature and higher productivity than the steppe areas under relative arid condition. The most arid areas on northwest QTP were generally overgrazed and with low stocking rate threshold, which partially reflected the interaction between the spatial variation of aridity and grazing pressure. Some studies pointed out that increasing aridity could exacerbate the negative effects of overgrazing on grassland ecosystem[37], and accelerate grassland degradation[38]. The high stocking rate locations in the QTP meadows area usually had more livestock, and therefore had increasing grazing activities since the

1980s (Fig. 2). Temporal patterns of stocking rate partially reflected the effects of management policies launched in different periods. Stocking rate on eastern grasslands of the QTP increased substantially in the 1980s, likely due to the household contract responsibility system[39]. In the 1990s and early 2000s, however, the stocking rate on the grasslands decreased, reflecting the launch of several strict ecological conservation programs[39,40]. Most recently, stocking rate on eastern grasslands has again increased[41].

Our analysis suggests that current stocking rate in most grasslands on the QTP is below the stocking rate threshold inducing grassland degradation, particularly in central areas, which indicated that grazing activities is still within sustainable level over these areas. Nevertheless, several areas in the north, south and northwest are currently over the stocking rate threshold, and our modeling suggests that they currently may have already become degraded under the actual grazing practices. Reducing the stocking rate to the threshold level would lengthen the time until degradation to 60 years in southern regions and 90 years in northeastern regions, though it would remain less than 15 years in northwestern regions. Management of northwestern grasslands could be intensified (such as fencing and grazing bans) to lengthen the time to degradation. Reducing stocking rate to the simulated threshold level in southern and northeastern regions could allow more time to develop management plans for the longer term. Ultimately, management plans to prevent unsustainable increases in stocking rate within areas where it currently falls below the threshold may benefit from the information presented here. The difference map between stocking rate and the stocking rate threshold can inform these efforts (Fig. 3).

The climate condition over grassland of QTP was found to be in a warming and humidification trend in both history and future period

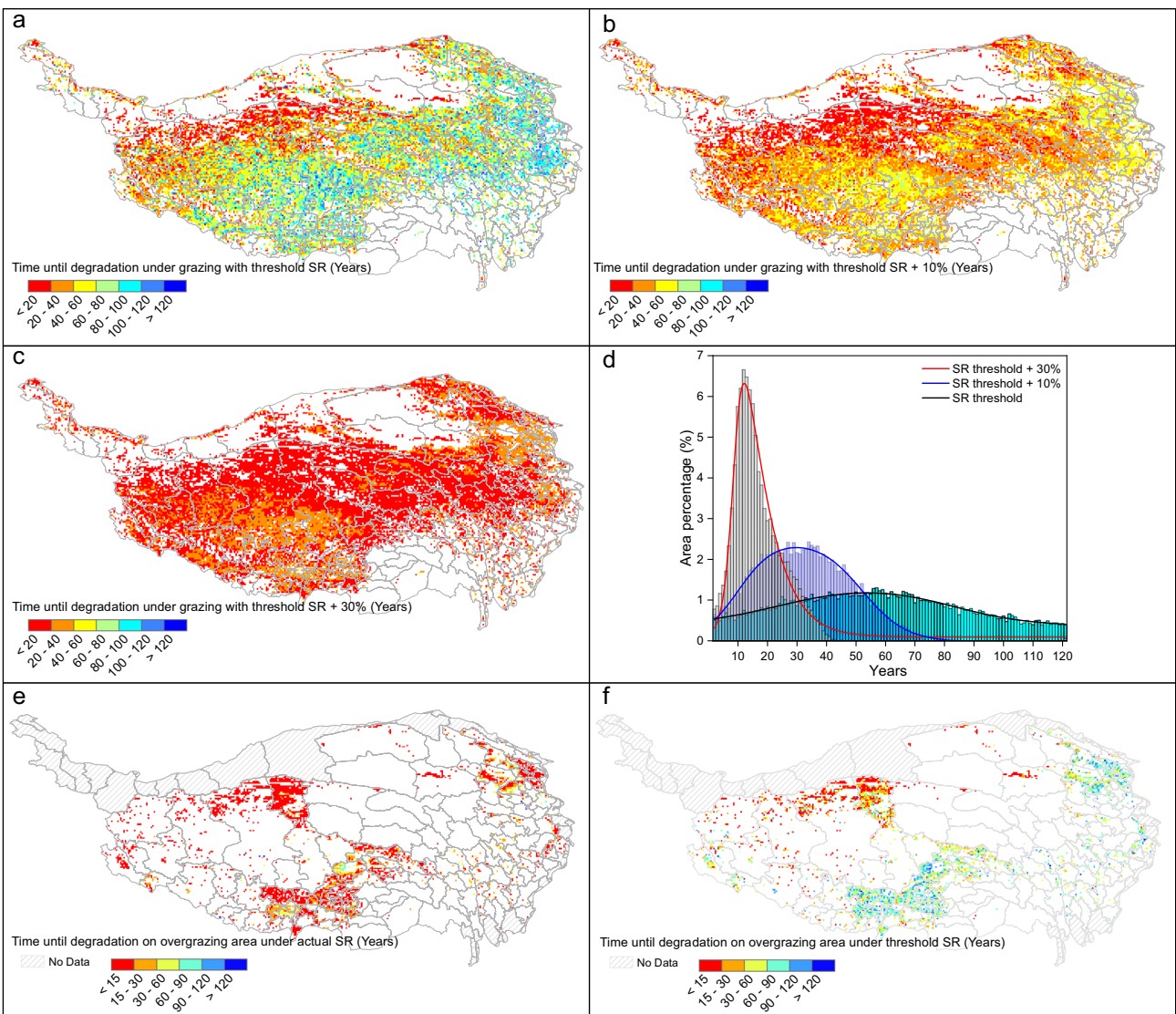

**Fig. 4 | Predicted time for grasslands on the Qinghai-Tibetan Plateau to become degraded, based on local stocking rate threshold and different stoking rates.** The colors refer to how many years until the area becomes degraded. Spatial patterns of time until degradation were examined assuming that stocking rate remained (**a**) at the threshold, (**b**) at 10% above the threshold, or (**c**) at 30% above the threshold. **d** Distribution of time until degradation in terms of amount of surface area. Spatial patterns of time until degradation for the actual overgrazed grassland area with stocking rate remained (**e**) at actual level, (**f**) at the threshold (extracted from **a**). Maps were masked with an initial grassland distribution as showed in Fig. 1. SR stocking rate.

(Supplementary Fig. 1, 2)[1,41]. Productivity of grassland would be enhanced under warming-wetting condition, and subsequently the carrying capacity of grassland, as well as the stocking rate threshold could be enhanced. Climate change and $CO_2$ fertilization together can partly offset the negative effects of grazing only over the areas with stocking rate below the threshold. Across the overgrazing grassland areas (with stocking rate exceeded threshold), the positive effect of climate change and elevated $CO_2$ was not enough to compensate the negative effects of grazing (Supplementary Note 1). The historical patterns further indicated that to keep a balance between livestock grazing demand and making full use of positive effects of climate change and elevated $CO_2$, stocking rate of about 60% (ranged from 50% to 70%) of the threshold could be appropriate. Therefore, although future stocking rate threshold would be enhanced with warming, humidification and elevated $CO_2$ concentration, especially in the south and east overgrazing areas (Fig. 6), current grazing policies still fall short in preventing grassland degradation. In overgrazed areas, stocking rate could be considered reducing below the threshold, including an outright ban in northwestern areas. The stocking rate

conducted with a value between 50% and 70% of the modeled stocking rate threshold could help for maintaining a balance between human demands and protection efforts in grasslands where stocking rate lies below the threshold. From an early warning signal of the stocking rate threshold determined by "time until degradation", an early implication was supposed to provide for grazing activities management on the QTP and preventing the grassland from degradation. Although we focused on stocking rate in this study, grazing management strategies need to consider additional factors affecting land degradation and productivity, such as the time of grazing and resting and the spatial distribution of herbivores[42].

For ensuring sustainable grazing across the entire QTP, our threshold approach based on simulations at the grid scale may prove more sophisticated and more effective in the long run, especially after further refinement, than traditional approaches based on large-scale inventories or local grazing data. Our stocking rate threshold grids can be extended backward in time to assess whether areas have historically been overgrazed, and they can be projected into the future to estimate grazing sustainability in the face of changing climate and elevated $CO_2$.

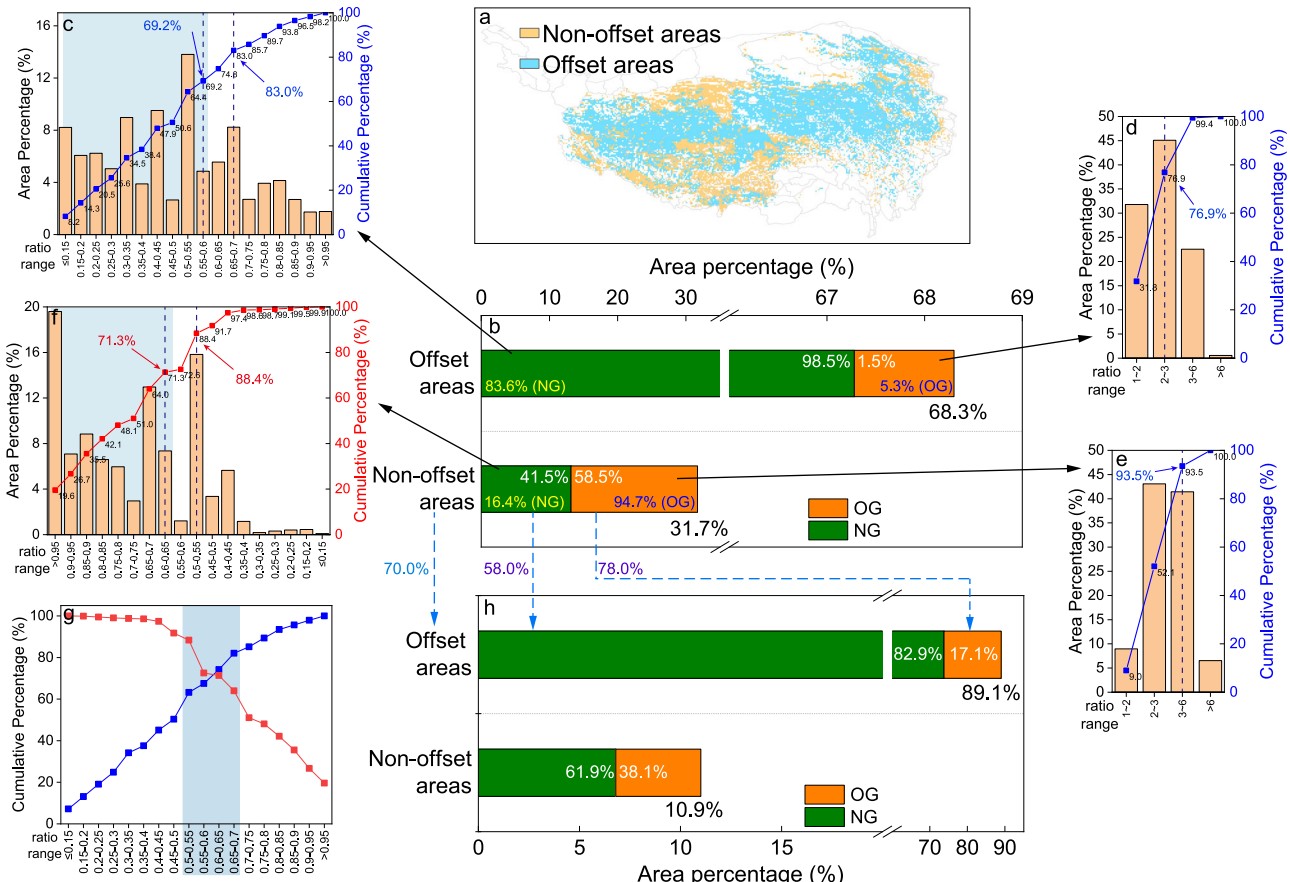

**Fig. 5 | Ratios of actual stocking rate to the stocking rate threshold on QTP grasslands. a** Distribution of areas where the negative effects of grazing are predicted to be offset by the positive effects of climate change and elevated $CO_2$ on grassland NPP ("offset areas") and areas where such offset is not predicted ("non-offset areas"); **b** The bar charts indicated the area percentage of "offset areas" and "non-offset areas" of the grassland on QTP, along with the area percentage (white numbers) of overgrazing (OG, orange color) and normal grazing (NG, green color) over each region, as well as the area percentage of OG (orange color; blue numbers) and NG (green color; yellow numbers) distributed in offset areas and non-offset areas respectively. The area percentage histogram and area percentage cumulative curve of the ratio of stocking rate to stocking rate threshold for the "offset areas" subject to NG (green color) (**c**) or OG (orange color) (**d**). The area percentage

histogram and area percentage cumulative curve of the ratio of stocking rate to stocking rate threshold for the "non-offset areas" of OG (orange color) (**e**) or NG (green color) (**f**), and different from others, the area percentage of stocking rate ratio in **f** accumulated decreasingly from 0.95 to 0.15. **g** The merged area percentage cumulative curves of the ratio of stocking rate to stocking rate threshold subject to NG (green color), extracted from **c** (blue line) and **f** (red line). **h** The bar charts indicate the area percentage of "offset areas" and "non-offset areas" of QTP grasslands, along with the area percentage (white numbers) of OG (orange color) or NG (green color) over each region based on an additional simulation that was performed in which the ratio of stocking rate to threshold was set to be 0.6 where the ratio was >0.6.

The spatial patterns of degradation, as well as the degradation criteria also need to be determined[43]. Our results may also facilitate the adoption of multi-dimensional interventions to optimize grazing patterns within specific areas[19], rather than a single "blunt force" intervention that may be effective only in certain regions[35,44].

Fencing, as one of the important grazing management measures, has been used widely on the QTP as part of pastoral land contracts or ecological protection projects to restore degraded grasslands[39,45,46]. While fencing appears to benefit the grassland ecosystem in the short term, its positive effects disappear after 6–8 years and it may even harm the ecosystem[46]. This has created substantial controversy around fencing[39,47], while our threshold approach may help to determine "when" and "where", as well as "how long" to establish fence over the plateau.

The stocking rate threshold calculated here take only grazing into account and therefore likely overestimate the "true" stocking rate threshold. Several human activities also affect grassland productivity significantly, such as urbanization, construction of roads and railways, reclamation and collection of herbal medicines[1,5]. Taking these factors into account could reduce the thresholds and likely shorten the time

until degradation. Indeed, in our analysis, increasing stocking rate by only 10% or 30% over threshold substantially shortened the time until degradation, particularly in northwest regions and overgrazed areas. In fact, the negative effects brought by other human activities mentioned above could be much larger than that induced by grazing activity with an increasing stocking rate of 10% to 30%. These results highlight the fragility of alpine grasslands. The threshold stocking rate should be considered as bottom line for ensuring the sustainability of grazing activities across grassland of QTP, particularly for its northwest part and overgrazing area.

Refinement of our method of calculating the stocking rate threshold for grassland degradation will require going beyond NPP to consider multiple indicators related to the plants and soil of the grassland ecosystem[10,11,43,48–50]. Meanwhile, the possibility of compensatory grass growth in response to light or moderate grazing was not included in the simulation[51], primarily because this possibility remains controversial[51,52]. Some studies have indicated that it is rare[53], or that it depends on the type and intensity of environmental stress factors[54] or nutrient availability[54,55]. Future studies could take compensatory growth into account and explore whether and how it may affect the

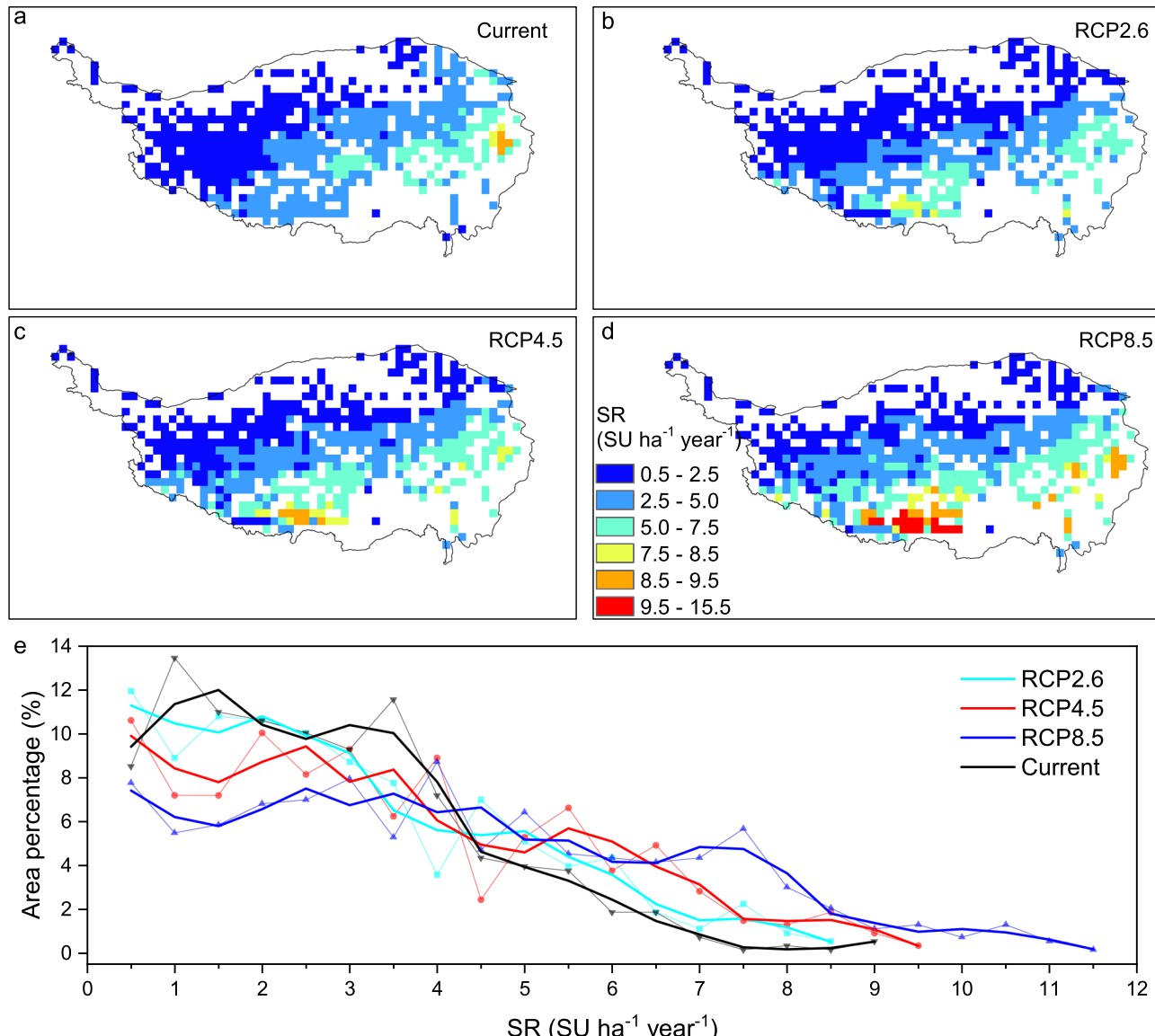

**Fig. 6 | Spatial patterns of stocking rate thresholds predicted under different climate conditions and CO₂ levels for period 2020–2100. a** Simulation based on current climate conditions and CO₂ level (2020) during period 2020–2100. **b–d** Simulations based on average climate conditions and CO₂ levels of period 2020-2100 under the indicated representative concentration pathway (RCP) scenarios (RCP2.6, 4.5, 8.5). **e** Distribution of stocking rate thresholds in terms of area percentage of surface under different scenarios. Dotted lines represent the original data, for which the solid line indicates a 5-year running average. Maps were masked with an initial grassland distribution as showed in Fig. 1. SR stocking rate, SU sheep unit.

overall grazing capacity of alpine grasslands as they undergo warming and humidification in the future. When estimating the stocking rate threshold, we also did not take into account fencing (including its spatial distribution and area) on the QTP[45,46], the effects of such fencing on plant productivity, or the overgrazing that can occur immediately outside fenced enclosures[46,56].

Refinement of our approach should also consider uncertainties in how grasses respond to increasing CO₂ concentration[57], and to potential constrains of nutrients (e.g. nutrient removal by grazing). Evaluating the individual contribution of different factors (e.g. precipitation, temperature and raising CO₂ concentration) to the thresholds also could make a better understanding in grassland ecosystem degradation mitigation under climate change conditions. Since climate change and grazing activities would affect the structure of palatable and unpalatable grasses, as well as the grazing tolerance, future model improvements could include detailed grass traits and species information instead of only two general grass plant functional

types (PFTs) (see Methods). Finally, the model performance evaluation should be improved by including more field data covering the entire QTP. These various refinements should make thresholds and the associated time until degradation more accurate for guiding management decision-making.

## Methods

### Underlying model

The TRIPLEX-GHG model is a dynamic global vegetation model that takes into account land surface processes, vegetation phenology, long-term vegetation dynamics, soil biogeochemical processes, greenhouse gas emissions, cycling of nitrogen and phosphorus, wetland area dynamics, and peatland carbon accumulation[58–60]. The TRIPLEX-GHG modules of vegetation dynamics, land surface, plant phenology, and soil biogeochemistry are based primarily on the Integrated Biosphere Simulator (IBIS) model[61,62] (Supplementary Fig. 3). The vegetation in TRIPLEX-GHG was represented by plant functional types (PFTs) and

two types of grasses (C3 and C4) were included. Vegetation dynamics was characterized by PFT changes in terms of biomass and leaf area index, with competition among PFTs for sunlight and water. Plant phenology was modeled based on the coldest temperatures, thresholds for accumulated growing degree days, and thresholds for 10-day average temperature. Canopy photosynthesis was modeled based on the Farquhar model to calculate gross primary productivity (GPP) and net primary productivity (NPP) for each PFT[61,62]. An improvement considering nitrogen constraints on estimation of canopy NPP was adopted by applying a soil N availability modifier and dynamic C:N ratios in order to capture how the carbon cycle responds to dynamic nitrogen availability[58,63].

For the present study, we focused on the PFT of grasses and integrated a grazing module into the model to estimate stocking rate threshold for detecting grassland degradation on the Qinghai-Tibetan Plateau (QTP) in the face of climate change and elevated $CO_2$. The parameters of the two grass PFTs in the original IBIS model[61] were adjusted to fit the natural vegetation distribution in China by Yuan et al.[64] (Supplementary Table 1).

## Grassland NPP

In this study, we adopted NPP as an indicator to detect degradation of QTP grasslands and to evaluate the effect of climate change, elevated $CO_2$ and grazing activities on the grasslands. NPP, the net carbon assimilated by plants and sensitive to climate change and human activities, is an important proxy index of vegetation growth status and ecosystem health. It has been proved to be a good estimator of ecosystem functioning[33] and land degradation[34]. NPP was calculated as the balance between the carbon gained by GPP and carbon released by plant maintenance respiration, as described by the equation in the IBIS model[61,62]:

$$NPP = (1 - \eta) \int \left(A_g - R_m\right) dt \qquad (1)$$

where $A_g$ was the gross photosynthesis rate; $R_m$, the sum of maintenance respiration rates; and $\eta$, the fraction of carbon lost due to growth respiration. Gross photosynthesis rate was calculated based on the Farquhar equations[65] and expressed as the smaller of the light-limited rate or Rubisco-limited rate of photosynthesis in the case of C3 plants, or as the smallest of the light-limited, Rubisco-limited or $CO_2$-limited rates of photosynthesis in the case of C4 plants[66]. For the calculation of Rubisco-limited rates of photosynthesis, the maximum carboxylation capacity of Rubisco was modified using the leaf C:N ratio in order to take into account potential nitrogen limitation[58,63]. For the calculation of productivity, a soil N availability modifier was applied[58,63].

## Grazing processes on QTP grasslands

We constructed the grassland grazing processes framework (Supplementary Fig. 3) based on several studies[67–70] and defined the summer grazing season as May to October, and the winter grazing season as November to April.

## Grassland biomass consumption

Grassland biomass consumption was calculated based on stocking rate and herbage mass availability. During the summer grazing season, the total biomass needs for livestock consuming ($BioGrz_t$, kg C m$^{-2}$) was calculated as[68]:

$$BioGrz_t = LS_w \times r_{IT} \times C_{coe}/10000.0 \qquad (2)$$

where $LS_w$ was livestock weight (kg ha$^{-1}$), which depends on stocking rate (sheep unit (SU), ha$^{-1}$ year$^{-1}$) and initial livestock unit body weight (40 kg); $r_{IT}$ was livestock intake rate, which was set to 0.02 based on the general intake rate of Tibetan-sheep[69] and an approximate daily

body weight gain of 40 g per sheep[69]; $C_{coe}$ was the unit conversion factor of dry mass to carbon (0.475); and 10000.0 served to convert from ha to m$^2$. An intake rate of 0.02 meant that Tibetan-sheep consumed biomass each day equivalent to 2% of body weight.

We assumed that in the summer grazing season, 95% of biomass consumption was live grass and 5% dead material. The aboveground grass biomass (AGB), live or dead, was assumed to be sufficient for grazing if it exceeded $BioGrz_t \times 0.95 \times 2.0$. The biomass consumed from live grass ($BioGrz_l$) was calculated in units of kg C m$^{-2}$ according to the study of Shiyomi et al. (2011)[68]. We added an exponential function to calculate biomass consumed from live grass, while AGB was allowed to range from $BioGrz_t \times 0.95$ to $BioGrz_t \times 0.95 \times 2.0$ in order to make biomass consumption vary continuously.

$$BioGrz_l = \begin{cases} BioGrz_t \times 0.95 \; when \, AGB > BioGrz_t \times 0.95 \times 2.0 \\ AGB \times \left(0.08 \times \exp\left(1.8326 \times \frac{AGB}{BioGrz_t \times 0.95 \times 2.0}\right)\right) \\ \quad when \, BioGrz_t \times 0.95 < AGB < BioGrz_t \times 0.95 \times 2.0 \\ AGB \times 0.2 \; when \, AGB < BioGrz_t \times 0.95 \end{cases} \qquad (3)$$

The biomass consumed from standing dead grass ($BioGrz_{d-SGS}$), which the model considered as part of litterfall, was calculated in units of kg C m$^{-2}$ as:

$$BioGrz_{d-SGS} = BioGrz_t \times 0.05 \qquad (4)$$

During the winter grazing season, we assumed the biomass needed for livestock come from dead plant material in the above-ground litter pool. In the winter grazing season, livestock was assumed to lose 25% of body weight[69], and this weight loss was distributed equally across all days of the winter grazing season. The daily biomass consumed by livestock was considered to compensate for the difference between livestock daily respiration consumption and body weight loss. Thus,

$$BioGrz_{d-WGS} = LS_w \times 0.012 - LS_{w-ESGS} \times 0.25/D_{WGS} \qquad (5)$$

where $BioGrz_{d-WGS}$ was the daily biomass consumed from standing dead grass in the winter grazing season; $LS_{w-ESGS}$, livestock body weight at the end of the summer grazing season; $D_{WGS}$, the number of days in the winter grazing season; and $LS_w$, livestock body weight, which was adjusted for daily weight loss at a proportion of 0.012 to body weight.

**Livestock weight dynamics.** Livestock body weight of ($LS_w$) was calculated in daily steps using the equation as:

$$LS_w = LS_w + \left(BioGrz_l \times 0.65 + BioGrz_d \times 0.45\right) \times 10000.0/C_{coe} - LS_w \times 0.012 \qquad (6)$$

where 0.65 and 0.45 represented the digestibility of live and dead plant material, respectively, while 0.012 represented the energy necessary for daily respiration consumption to maintain livestock activity. Indigestible material from live and dead plants was assumed to be excreted by animals at respective excretion rates of 0.35 and 0.55 and to return to the soil pool, where it participated in soil biogeochemical processes.

We assumed the negative effect of trampling on above-ground biomass depend on stocking rate with an effect coefficient of 0.8%[67,70].

## Mapping stocking rate on QTP grasslands

To map stocking rate across QTP grasslands, we collected livestock (including sheep, goat, pig, cattle, yak, horse, mule, donkey, camel etc.) numbers for each province or county across the QTP from provincial or prefecture statistical yearbooks for the period from 1980 to 2017 (National Digital Library of China, http://www.nlc.cn/). We converted livestock numbers into sheep units (SUs) as described in Ren[69]

(Supplementary Table 2). All data processing and modelling steps related to grazing activity in this study were based on SUs.

We calculated annual average stocking rate for each county as:

$$SR_i = \frac{1}{2} * \left( \frac{SU_i}{A_i * R * S} + \frac{SU_i}{A_i * R * W} \right) \quad (7)$$

where $SR_i$ was the stocking rate in county $i$ (SU ha$^{-1}$ year$^{-1}$); $SU_i$, SUs in county $i$ (capita); $A_i$, the area of grassland in county $i$ (ha); $R$, the area fraction of grassland in county $i$ that was edible; $S$, the area fraction of summer pasture in county $i$; and $W$, the area fraction of winter pasture in county $i$. To generate dynamic maps of annual stocking rate at the county level, maps of annual grassland distribution and area were retrieved from the ESA CCI Land Cover time-series dataset (http://www.esa-landcover-cci.org/). The area fractions of edible grassland, summer pasture, and winter pasture for each county were collected from provincial or prefecture statistical yearbooks. Since these data were unavailable on an annual basis, we calculated average values over the years for which data were available. Finally, dynamic maps of stocking rate at the county level were calculated for the period from 1980 to 2017.

### Data for model performance evaluation

The modeled stocking rates were compared with designed heavy level of stocking rates in local grazing experiments collected from 38 field sites (Supplementary Fig. 4), and the stocking rate at each site was classified as heavy, medium, or light. We also compared the observed biomass, which were available for 15 sites (Supplementary Fig. 4, Supplementary Table 3), with the modeled biomass in the corresponding grid under different stocking rates.

We compared simulated and eddy covariance retrieved monthly GPP for three flux sites (CN-Dan, CN-Ha2, CN-HaM) involved in FLUX-NET Network (https://fluxnet.org/) that located on the QTP to validate the GPP simulation performance of the model (Supplementary Fig. 4).

We also compared the modeled NPP and NPP productions retrieved from Moderate Resolution Imaging Spectroradiometer (MODIS) (http://files.ntsg.umt.edu/data/NTSG_Products/MOD17/GeoTIFF/MOD17A3/GeoTIFF_30arcsec/) and Advanced Very High Resolution Radiometer (AVHRR) (http://www.glass.umd.edu/NPP/AVHRR/) to validate the NPP simulation performance of the model at grid scale.

### Model forcing data

Daily meteorological data on precipitation; mean, maximum, and minimum temperatures; relative humidity; solar radiation, and wind speed were generated for whole China between 1960 and 2017 by smooth thin plate spline interpolation[71] at a resolution of 0.08333° based on observed data from 2400 national meteorological stations. Then, daily climate data were extracted from this national dataset for the model simulations on the QTP. Soil texture on the QTP was derived from a high-resolution soil texture map of China[72], and initial grassland distribution information was derived from the 1:1000000 China vegetation map[73]. All data were generated with spatial resolution of 0.08333° (~10 km).

### Model simulation

The model was firstly set up as a 400-year spin-up simulation with multiyear average historical meteorological data for the period between 1960 and 1990. This allowed the ecosystem carbon pools, especially soil carbon pools, to reach equilibrium. For the simulations to evaluate the effects of climate change, elevated $CO_2$ concentration and stocking rate on NPP of QTP grasslands (Supplementary Note 1), the model ran with daily meteorological data beginning from 1960 to 2017, while grazing activity was applied from 1980 to 2017. Grazing

activities on the QTP were applied in the simulations from 1980 (by using the generated stocking rate maps), since this marked the beginning of livestock privatization and dissolution of the collective system on the Plateau, from when many pastoralists increased their herd sizes under a condition of rapid transition to a market economy with exacerbating the trend toward high livestock numbers[8,40]. Grassland productivity, biomass removed by grazing, and livestock body weight were updated using a daily time step in the model.

### Stocking rate threshold detection

We assumed that grassland productivity would decrease with increasing stocking rate. The consumption of NPP ($NPP_{Grz}$) at each daily timestep was calculated as the minimum of biomass consumed from live grass ($BioGrz_l$) that directly depended on stocking rate and the NPP could be allocated to leaf:

$$NPP_{Grz} = \min(NPP_D * \alpha_{leaf}, BioGrz_l) \quad (8)$$

where $NPP_D$ was daily net primary productivity of grassland (kg C m$^{-2}$), $\alpha_{leaf}$ was the allocation fraction of total photosynthate to leaf (Supplementary Table 1).

In order to detect the stocking rate threshold for grassland degradation in each grid cell, the model was run under different stocking rates, from non-grazing (stocking rate = 0.0) to maximum possible grazing (stocking rate = 15 SU ha$^{-1}$ year$^{-1}$), increasing in a step of 0.5 SU ha$^{-1}$ year$^{-1}$ for a period of 120 years from the grazing activities applied on (Supplementary Fig. 5). During this period, multiyear (1980-2017) averages of daily meteorological and $CO_2$ concentration data were used to drive the model in order to eliminate the effects of varied climate and ramping $CO_2$ concentration on NPP in the process of threshold detection. Altogether 31 simulations were performed over the entire QTP. When the NPP of a cell reached 1% of the baseline NPP before the stocking rate was applied, we defined that stocking rate to be the stocking rate threshold for the grid cell, above which stocking rate would lead to degradation based on the indicator of NPP (Supplementary Fig. 5). This simulation strategy was performed on each grassland grid cell (~10 km) of the QTP. Using this approach, and 3720 annual NPP maps (31 scenarios × 120 years) of grassland on the QTP were used to generate a map of stocking rate threshold and a map showing how many years before the grassland in each grid cell at the indicated stocking rate ("time until degradation"). These simulations under different stocking rates were run with average climate and $CO_2$ conditions between 1980 and 2017.

Using a similar approach, we generated stocking rate threshold maps under future climate change. The modeling was run using stocking rates ranging from non-grazing (stocking rate = 0.0) to maximum possible grazing (stocking rate = 15 SU ha$^{-1}$ year$^{-1}$) by increasing the grazing level in a step of 0.5 SU ha$^{-1}$ year$^{-1}$ under the scenarios of RCP 2.6, 4.5 and 8.5, corresponding to low, medium, or high emissions in the 21st century (Supplementary Table 4). Multiyear (2020–2100) averaged climate and $CO_2$ concentration data for each scenario was used to drive the model. A fourth scenario was considered in which climate conditions and $CO_2$ concentration were held constant at their 2020 levels. A total of 31 simulations across QTP grasslands were performed for each scenario, resulting in stocking rate threshold maps. Model driving data on precipitation, temperature, relative humidity, and radiation of three RCPs were averaged from 24 General Circulation Models (GCMs) at a spatial resolution of 0.5° (Supplementary Table 5).

### Model performance evaluation

We make a comparison between the heavy grazing levels designed in the field experiment and stocking rate thresholds simulated in this study as a reference to evaluate the modeled stocking rate threshold

(Supplementary Fig. 4). Modeled stocking rates were compared at the county level with "heavy" stocking rates at 38 field sites (Supplementary Fig. 4, Supplementary Table 3) in local grazing experiments in which sites were subjected to "light", "medium" or "heavy" stocking rate. Although this comparison may be imperfect because stocking rates defined as "heavy" in the literature have an unclear relation to stocking rate thresholds defined in our study, regional mean simulated stocking rate thresholds were quite consistent with designed heavy stocking rates (slope = 0.923, $R^2 = 0.77$, $p < 0.01$; Supplementary Fig. 6). Generally, the heavy stocking rates in experimental studies were slightly greater than the simulated stocking rate thresholds (Supplementary Fig. 6). The difference in spatial scale between our modeling grid (~10 km) and local grazing experiment site (~several hectares) meant that the same modeling grid sometimes contained multiple sites differing substantially in stocking rates designated as heavy in experimental studies (Supplementary Table 3).

We also compared simulated and measured AGB at the sites with different stocking rates for which data were available (Supplementary Table 3). Comparisons were performed on a county scale (grouped the site with county location) (Supplementary Fig. 4, Supplementary Table 3). The two sets of values were generally comparable and showed the gradient influence of stocking rate on AGB (Supplementary Fig. 7). Agreement between simulated and observed AGB values was acceptable across all five regions ($R^2 = 0.74$, $p < 0.01$; Supplementary Fig. 7).

The simulated GPP values agreed well with the eddy covariance retrieved GPP for the three sites with coefficient of determination all greater than 0.74 ($p < 0.001$) (Supplementary Fig. 8). The discrepancy of GPP between simulated and observed data may be caused by the inconsistent spatial scale between modelling grid (~10 km) and footprint of flux tower.

Multi-year average NPP (2000-2015) was compared between simulated in this study and the productions retrieved from remote sensed data of AVHRR and MODIS at grid scale across QTP (Supplementary Fig. 9) for a reference of model performance evaluation. The simulated NPP values agreed better with the AVHRR data ($R^2 = 0.62$, Supplementary Fig. 9b) than with the MODIS data ($R^2 = 0.47$, Supplementary Fig. 9a). Simulated NPP values showed a coefficient of determination of nearly 0.6 when compared with NPP values averaged between the MODIS and AVHRR datasets (Supplementary Fig. 9c). The AVHRR data usually contained greater NPP values than the MODIS data (Supplementary Fig. 9d).

### Reporting summary
Further information on research design is available in the Nature Portfolio Reporting Summary linked to this article.

## Data availability
The source data underlying Figs. 1–6 are provided as Source Data files and have been deposited in the Figshare database (https://doi.org/10.6084/m9.figshare.24191823). Daily meteorological data of national meteorological stations were obtained from China Meteorological Data Service Centre (https://data.cma.cn/) and the interpolated meteorological driving data is publicly available at National Earth System Science Data Center (http://www.geodata.cn/). ESA CCI Land Cover time-series dataset can be obtained at http://maps.elie.ucl.ac.be/CCI/viewer/download.php. The monthly GPP dataset for three flux sites on QTP is publicly available at FLUXNET Network (https://fluxnet.org/data/download-data/). The NPP productions retrieved from Moderate Resolution Imaging Spectroradiometer (MODIS) is available at: http://files.ntsg.umt.edu/data/NTSG_Products/MOD17/GeoTIFF/MOD17A3/GeoTIFF_30arcsec/. The NPP productions retrieved from Advanced Very High Resolution Radiometer (AVHRR) is available at the link: http://www.glass.umd.edu/NPP/AVHRR/. Source data are provided with this paper.

## Code availability
Custom R scripts for stocking rate threshold detection and time until degradation detection are available in the Figshare database (https://doi.org/10.6084/m9.figshare.24191823). The source code of the terrestrial ecosystem model of TRIPLEX-GHG is available by contacting the corresponding authors (zhuq@hhu.edu.cn; yfwang@ucas.ac.cn).

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

## Acknowledgements

This work was financially supported by the Second Tibetan Plateau Scientific Expedition (Grant No. 2019QZKK0304-02, Y.W., Q.Z. and H.C.), the National Natural Science Foundation of China (Grant Nos. 42041005, Y.W. & U2243203, LR), and the National Key R&D Program of China (Grant No. 2016YFC0501804, Y.W. and Q.Z.). J.L.'s work was performed under the USGS Climate Research and Development Program. Acknowledgement for the data support from "National Earth System Science Data Center, National Science & Technology Infrastructure of China (http://www.geodata.cn)". Nonendorsement disclaimer: Any use of trade, firm, or product names is for descriptive purposes only and does not imply endorsement by the U.S. Government.

## Author contributions

Q.Z. and H.C. designed research, performed research, analyzed data, wrote the paper. Y.W. designed research, analyzed data, and revised the manuscript. C.P., S.P., J.H., S.W., X.Z., Q.Y., L.R. discussed the results and revised the manuscript. J.L. and J.Z. contributed new analytic tools and analyzed data. Q.Z. and J.L. improved the model code and conducted the simulations. J.Z., X.F., J.J. performed research and analyzed data.

## Competing interests

The authors declare no competing interests.
