## [Peer Review File · Nature Communications]

nature portfolio

Peer Review FileReviewer comments, first round:

Reviewer #1 (Remarks to the Author):

The authors define grazing capacities for areas of the Qinghai-Tibetan Plateau (QTP) using a process-based ecosystem model. They then compare those capacities against recent stocking rates and speak of stocking relative to a threshold they have identified. Climate projections are then used to infer changes into the future. The writing is of high quality, although the work would benefit from additional edits. The analyses too are interesting. I have several overarching comments, and a few more minor points.

I was quite excited and impressed when first reviewing the manuscript to learn that the authors had identified thresholds in grazing intensity in QTP. The definition of quantitative thresholds in sustainability science is notoriously difficult. But I think this is related to the multiple meanings of "threshold". The exciting, and I think more typical, interpretation in sustainability science is of a quantitative measure of use over which a non-linear response will be seen, leading to a dramatic decline in resource availability. Many examples, a classic being the rapid collapse of Atlantic cod fisheries in the northwestern Atlantic in 1992. Identifying those thresholds a priori is extraordinarily challenging and exciting. Imagine, as the authors' title suggests, that early warning signals for approaching grazing intensity thresholds had been identified (as suggested on lines 68 and 70). That would be a great advancement. In reading the work I find that threshold is used in another context, to speak of recommended limits not to be exceeded. That is useful, but typical. I worry that the authors will add confusion by using "threshold" where the long-standing rangeland term "carrying capacity" is instead intended. I would suggest the authors consider alternative language to describe their recommendations, avoiding the term that may cause confusion. Put another way, lines 251-252 say "... our threshold approach may prove a more sophisticated and, in the long run, more effective way to ensure sustainable grazing." But how is this approach distinct from other explorations of grazing carrying capacities, using simulation or otherwise?

The authors fail to compare their results to the works of many others who have mapped global grassland degradation (e.g., Yang 1992; Wu et al. 2009; Ho-Young et al. 2015; Cao et al. 2019; Bardgett et al. 2021; Sun et al. 2022; Zhou et al. 2023, recognizing some are too new to be considered). Some of these works either include figures that show degradation in the QTP or are specific to the QTP. The portion of QTP degraded includes a broad range of estimates, some as high as 50%. To be clear, the authors cite degradation of the QTP, but do not place their grazing threshold in the context of the remainder of the literature. The authors say that 80% of the QTP is below some threshold stocking level. But they also say they seek to define "the stress that grassland could take for supporting productivity before a degradation condition is triggered" (lines 74-75). That suggests that the authors view at least 80% of the QTP as undegraded. Comparing the contrasting finding to that reported in the literature would improve the manuscript. Perhaps defining the threshold cited (related to the previous point) and how it relates to degradation, if it does, would be helpful. The authors then calculate (lines 137-143 and many other locations) how long areas with grazing exceeding their threshold would take to become degraded. That suggests that the authors see only some small fraction of the 20% of land exceeding the threshold as currently degraded. Contradictions in this regard in the writing itself and in the writing compared to the literature exist and should be rectified.

I invite the authors to consider the section starting at line 158 for review. It is not as clear as the earlier material. I think it is because the actual grazing intensity and threshold intensities inform statistics, and so two percentages are often provided, which is a little confusing. So "... nearly 89% and 72% of the GI ratio larger than 0.5 and 0.6 under normal grazing condition, respectively" and "... to offset the negative effects of grazing with area percentage of 72% or 89% ..." is a little tough going for readers. The same information may be shared in almost as pithy a way while untying the combined statements.

Some observed grazing rates converted to categories were used in assessment, which is helpful. It seems there remains a great deal of remotely sensed information amenable to these broad-scale

analyses that is suitable for simulation assessment as well, and readily available. Typical data used for a model that has NPP estimates at its core would be NPP and GPP estimates, plus NDVI or EVI, from MODIS, for example. The authors may wish to describe why those data were not used in assessment or incorporate their use.

Line 33. There is ambiguity in the 70% figure. Is it 70% of the area in question? Or 70% of the effect offset? Something else?

Line 48. Ensuring the "ecological security of the QTP" may be briefly expanded upon. I'm curious, anyway.

Line 102. "Processed model" sounds a little unusual. Perhaps "process-based model"?

Line 112. Briefly define sheep units for those working in other regions.

Lines 163-164. "To some extent" may be removed given that "partly" was used.

Line 167. GI is introduced without having been defined.

Lines 251-252 introduce a comparison without the thing being compared against. It speaks of a more effective way, compared to ... I'm not speaking of rangeland science here, only the writing. "compared to traditional approaches" would be sufficient, for example.

May lines 413-415 ("When the NPP of a cell reached nearly to zero under a certain grazing intensity, then we defined that grazing intensity to be the threshold value for the grid cell, such that intensities above that value would lead to degradation.") somehow find their way into the body of the manuscript? This is really the heart of the matter, the means in which you defined a threshold value (the confusing name aside). A summary of this approach in the Introduction or Results would be very helpful, and require just a few words (e.g., a threshold was crossed when NPP was < 0 [but using another word for "threshold"]).

Reviewer #2 (Remarks to the Author):

The manuscript entitled "Early warning of grazing-induced thresholds of grassland degradation on the Qinghai-Tibetan Plateau", by Zhu et al. addresses the impact of grazing intensity on net primary productivity (NPP) along the Qinghai-Tibetan Plateau (QTP) grassland ecosystems, using a process-based ecosystem modeling. The particular objectives were (i) to determine the grazing intensities that induce an abrupt reduction in NPP, (ii) to apply these findings to identify grassland areas under threat of degradation, and (iii) to predict when such degradation might occur under current and future grazing and climate conditions. The main results indicate that most grasslands of the QTP are managed with grazing intensities below those that induce the abrupt reduction in NPP, and that maintaining the grazing intensity around 60% of that induces the abrupt change has the potential to sustain the human demand without inducing grassland degradation. These aspects are critical since this extensive area is mostly covered by grasslands that sustain millions of domestic herbivores, which in turn are the livelihoods of millions of people.

The manuscript is clear and the study is well performed. The findings are novel and have the potential to generate some international interest in its field of study. However, there are some issues that should be addressed. My main concern is the unique ecological attribute and the relationship to estimating thresholds (grazing intensity and NPP), which can lead to misunderstanding degradation processes since the study did not consider other critical ecological attributes and other aspects of grazing management beyond the stocking rate (here grazing intensity). On the one hand, degradation processes can be evidenced through ecosystem attributes different from NPP, which can be more appropriate to early detect degradation (e.g. Kéfi et al., 2007; Berdugo et al., 2017, 2020; Oñatibia & Aguiar, 2023). For example, degradation may occur without inducing changes in NPP (see Verón et al., 2006). On the other hand, grazing management strategies are much more complex than grazing intensity. For example, grazing and

rest periods and the spatial distribution of herbivores in the landscape are key factors to control land degradation and promote productivity (see e.g. Briske, 2017). I suggest the authors expand the section on potential limitations of the study including a discussion of these issues. Besides, both contextualizing the study and comparing the main findings with existing global literature (other than just for the QTP) on early warning signals of degradation and ecosystem thresholds are also recommended.

Regarding the model validation, the field sites are relatively few and cover a small area compared to the large extent of the whole study area. This weakness is not fatal, but the authors should discuss this aspect, maybe also expanding the section about the potential limitations of the study. Finally, a minor issue is that the treatment given to the term "threshold" is vague and sometimes unclear (e.g. lines 67, 74, 78, 88, 102, and see specific comments below). This is important since a central role is given to this term throughout the manuscript. In general, it is not completely clear how it relates to an early warning signal for grassland degradation. Definitions of threshold and degradation (and their relationship) from the introduction will help to understand the main ideas addressed. A more precise alternative could be directly referring to grazing-induced abrupt NPP reduction instead of threshold and degradation.

SPECIFIC COMMENTS

Lines 1-2. I suggest revisiting the title after considering the comments on what thresholds represent and how they are related to early warning signals of grassland degradation.

Lines 26-27. Suitable threshold? Threshold of what? Shouldn't it be "suitable grazing intensity"?

Lines 50-51. The authors indicate that QTP grasslands are very fragile. Please, justify this statement and add citations. If this information is that found in lines 64-66, please, avoid repetition.

Lines 58-62. Please define degradation. A definition of what grassland degradation represents to the authors would help make this section clearer.

Line 140. Remove "that".

Lines 220-227. Areas described as overgrazed seem to be generally the most arid (e.g. northwest). If this is correct, authors can discuss the potential interaction between increasing grazing and aridity (see e.g. Gaitán et al., 2018; Oñatibia et al., 2020).

Lines 366-368. If grazing intensity is the annual average, its units could be SU/ha/year (stocking rate units) instead of density units (SU/ha).

Line 425. I suggest moving this Figure with an expanded legend to the main manuscript.

Line 608-609. I suggest that authors indicate what the white color represents on the map; if it means the absence of data or if represents other vegetation types different from steppes and meadows.

Figure legends. Information is generally lacking for the figures to be self-explanatory, particularly those in the supplementary material.

References

- Berdugo, M., Kéfi, S., Soliveres, S., & Maestre, F. T. (2017). Plant spatial patterns identify alternative ecosystem multifunctionality states in global drylands. *Nature ecology & evolution*, 1(2), 0003.
- Berdugo, M., Delgado-Baquerizo, M., Soliveres, S., Hernández-Clemente, R., Zhao, Y., Gaitán, J. J., ... & Maestre, F. T. (2020). Global ecosystem thresholds driven by aridity. *Science*, 367(6479), 787-790.
- Briske, D. D. (2017). *Rangeland systems: processes, management and challenges* (p. 661). Springer Nature.
- Gaitán, J. J., Bran, D. E., Oliva, G. E., Aguiar, M. R., Buono, G. G., Ferrante, D., ... & Maestre, F. T. (2018). Aridity and overgrazing have convergent effects on ecosystem structure and functioning in Patagonian rangelands. *Land Degradation & Development*, 29(2), 210-218.
- Kéfi, S., Rietkerk, M., Alados, C. L., Pueyo, Y., Papanastasis, V. P., ElAich, A., & De Ruiter, P. C. (2007). Spatial vegetation patterns and imminent desertification in Mediterranean arid ecosystems. *Nature*, 449(7159), 213-217.
- Oñatibia, G. R., Amengual, G., Boyero, L., & Aguiar, M. R. (2020). Aridity exacerbates grazing-induced rangeland degradation: A population approach for dominant grasses. *Journal of Applied Ecology*, 57(10), 1999-2009.
- Oñatibia, G. R., & Aguiar, M. R. (2023). On the early warning signal of degradation in drylands: Patches or plants? *Journal of Ecology*, 111(2), 428-435.

Verón, S. R., Paruelo, J. M., & Oesterheld, M. (2006). Assessing desertification. *Journal of arid environments*, 66(4), 751-763.

Reviewer #3 (Remarks to the Author):

The manuscript "Early warning of grazing-induced thresholds of grassland degradation on the Qinghai-Tibetan Plateau" by Zhu & Chen et al. attempts to identify grazing intensity thresholds that would lead to grassland degradation. The question is of broad relevance, and such results could potentially be used as the basis for future grazing management in the area.

Overall the manuscript did not provide enough information for me to assess the results and conclusions drawn critically. The model, in particular the vegetation dynamical component, is not sufficiently described in either the main text or supplementary materials to allow appropriate assessment of the presented results and conclusions. The authors refer to two published manuscripts (Zhang et al., 2017; Zhu et al., 2014). I have endeavoured to follow the cited manuscripts to gain a better understanding of the model used but have done so unsuccessfully.

In Zhu et al., (2014), the model description leads to a string of further citations: "The TRIPLEX-GHG model (Peng et al., 2013) is based on the legacy of well-established and published models that include IBIS (Foley et al., 1996), DNDC (Li, 2000), TRIPLEX (Peng et al., 2002) and CASACNP (Wang et al., 2010). ". The cited Peng et al. (2013) paper does not appear to exist. At best, I can find an abstract submitted to the European geoscience Union (EGU), which is associated with TRIPLEX-GHG. This manuscript, however, indicates that IBIS represents the vegetation model component.

Zhang et al. (2017) explicitly cite the above-mentioned EGU abstract (Integrating greenhouse gas emission processes into a dynamic global vegetation model of TRIPLEX-GHG) at the beginning of the description of the model. Model improvements by (Liu et al., 2005) are also mentioned here. Based on the equation for NPP provided by the authors (line 303), I am guessing that the model version they are using is not that developed by Liu et al. (2005) and does not include any nutrient constraints on growth or photosynthesis. This is guesswork, though, and I did not see any code submitted, which could have been checked.

One of the major results presented by Zhu & Chen et al. is that the positive effects of climate change can offset, to a large extent, grazing. Based on the evidence presented, it is not clear whether such positive effects are as a result of temperature, precipitation, or increasing CO₂. Presumably (this is not stated), the authors have turned off all other plant functional types other than temperate C₃ grasses or similar for the present study. The extent to which CO₂ may fertilise C₃ grasses is uncertain (Reich et al., 2018). It is not clear what IBIS version the authors used and whether this version limits photosynthetic rate via any mechanism, e.g. nutrient limitation. There are no future change simulations such as those detailed in Tab.S1.1 that include grazing, and changing climate, but have CO₂ fixed. Such simulations would have allowed assessment of the magnitude to which the CO₂ fertilisation effect was responsible for the positive effects of climate change. Further, potential interactions between nutrients, plant growth and photosynthesis, and nutrient removal via grazing do not appear to be included in the model.

Additionally, the grazing threshold used (NPP close to zero) is not defined well. I remain uncertain about what "close" means. This threshold would need to be accurately defined and extensively justified as it is a pivotal metric in the manuscript.

Overall the authors attempt to answer a highly relevant question, but the model (perhaps) used is not yet developed to the extent to sufficiently answer it. In any case, the authors have not provided enough information for me to critically assess the methodology.

Minor comments:

There was a general lack of transparency in the description of model and methods.

Text size often changes within sentences and at many points in the manuscript e.g. lines 71-72.

There were many grammatical and language errors which could have been easily identified prior to submission. I won't list all of these as there are tools available that could be used to flag such errors.

Line 61 - "leading to degradation" a concise definition of degradation is necessary.

Line 63 - "In parallel, the productivity of alpine grasslands has fallen by 30% since late 1990s 8, 12 ". The studies cited here are in relation to the QTP and the authors here indicate that productivity has decreased since the 1990s. The results presented by the authors (S1.1 a) indicate that productivity (NPP) has increased over the majority of the QTP for this approximate time period based on their simulations. I may have missed something; how do the authors explain this apparent contradiction?

Lines 91-100 Numerous changes in text size.

Lines 91-100 This paragraph sets the reasoning for the present study. The issue mentioned is that it is not clear whether local measurements can be extrapolated to regional scales. The authors should provide more detail about how their modelling approach can do this better and be more explicit about which processes they expect to be able to represent better with their approach.

Lines 101-106 Numerous changes in text size.

Lines 101-106 Given that one of the main objectives of the study was to define a grazing intensity threshold, it would be good to have this threshold defined and justified here. It remains uncertain to me exactly how this threshold is defined and what its justification is.

Future change scenarios: "Spatial pattern of future grazing intensity thresholds on QTP grasslands " (line 186) & Fig. 6 (line 646) - to what extent are the simulated increases in grazing intensity thresholds driven by increasing CO₂ alone?

Figure 5 is very difficult to understand.

Lines 434 - 445 It is difficult to understand. It is not clear to me what "When this comparison was repeated at the regional level after clustering sites by location" means and what was done. It is apparent from Fig. S2.6 Tab. S2.2 that the simulated grazing intensity threshold is almost always above the grazing level at the site, ca. 25/38. The authors write that potential causes for this may be the inconsistent spatial scale between the modelling grid (10km * 10km) and the grazing experimental site. Does the model simulate a 10x10 km patch of vegetation, or are they simulating something smaller (e.g. 1 Ha) and assuming this is representative of a 10x10km grid? What was the spatial scale of the site-level grazing experiments? The authors also mention that the site-level grazing experiments may have a grazing intensity that is in excess of the intensity threshold. How do they justify this? There are presumably measurements collected at these sites, e.g. grass growth (NPP), that could be used to justify this assumption. Based on the threshold methodology of the authors, wouldn't exceeding this threshold imply grass NPP at these sites is zero? Have the authors considered the alternative possibility that their model is wrong?

Supplement

Fig. S2.6 - what is a) and what is b)? This is not specified.

Fig. S2.7 - the difference between observed and simulated grass biomass seems rather large, up to 300% at some sites and grazing intensities.

The "biomass for grazing" equations line 330/331 did not appear correctly. Perhaps this is just a rendering issue, though.

Grazing intensity threshold with NPP close to zero - based on the NPP equation (line 303), wouldn't zero NPP imply GPP was equal to maintenance respiration and there is no growth? Detailed justification for the choice of threshold is necessary as this is crucial to the manuscript. At

what time-step is grass growth calculated? At what time-step grass biomass removed by grazing?
I can't figure out whether this happens at daily, monthly, or yearly time steps.

Liu, J. X., Price, D. T., & Chen, J. A. (2005). Nitrogen controls on ecosystem carbon sequestration: A model implementation and application to Saskatchewan, Canada. *ECOLOGICAL MODELLING*, 186(2), 178–195.

Reich, P. B., Hobbie, S. E., Lee, T. D., & Pastore, M. A. (2018). Unexpected reversal of C3 versus C4 grass response to elevated CO₂ during a 20-year field experiment. *Science*, 360(6386), 317–320. <https://doi.org/10.1126/science.aas9313>

Zhang, K., Peng, C., Wang, M., Zhou, X., Li, M., Wang, K., Ding, J., & Zhu, Q. (2017). Process-based TRIPLEX-GHG model for simulating N₂O emissions from global forests and grasslands: Model development and evaluation. *Journal of Advances in Modeling Earth Systems*, 9(5), 2079–2102. <https://doi.org/10.1002/2017MS000934>

Zhu, Q., Liu, J., Peng, C., Chen, H., Fang, X., Jiang, H., Yang, G., Zhu, D., Wang, W., & Zhou, X. (2014). Modelling methane emissions from natural wetlands by development and application of the TRIPLEX-GHG model. *Geoscientific Model Development*, 7(3), 981–999. <https://doi.org/10.5194/gmd-7-981-2014>

RESPONSES TO REVIEWERS' COMMENTS

Reviewer #1 (Remarks to the Author):

The authors define grazing capacities for areas of the Qinghai-Tibetan Plateau (QTP) using a process-based ecosystem model. They then compare those capacities against recent stocking rates and speak of stocking relative to a threshold they have identified. Climate projections are then used to infer changes into the future. The writing is of high quality, although the work would benefit from additional edits. The analyses too are interesting. I have several overarching comments, and a few more minor points. I was quite excited and impressed when first reviewing the manuscript to learn that the authors had identified thresholds in grazing intensity in QTP. The definition of quantitative thresholds in sustainability science is notoriously difficult. But I think this is related to the multiple meanings of “threshold”. The exciting, and I think more typical, interpretation in sustainability science is of a quantitative measure of use over which a non-linear response will be seen, leading to a dramatic decline in resource availability. Many examples, a classic being the rapid collapse of Atlantic cod fisheries in the northwestern Atlantic in 1992. Identifying those thresholds a priori is extraordinarily challenging and exciting. Imagine, as the authors’ title suggests, that early warning signals for approaching grazing intensity thresholds had been identified (as suggested on lines 68 and 70). That would be a great advancement.

RE: Thanks for your constructive comments and positive feedbacks. We have tried our best to improve the quality of MS based on your comments and suggestions! The responses are listed below point by point.

In reading the work I find that threshold is used in another context, to speak of recommended limits not to be exceeded. That is useful, but typical. I worry that the authors will add confusion by using “threshold” where the long-standing rangeland term “carrying capacity” is instead intended. I would suggest the authors consider alternative language to describe their recommendations, avoiding the term that may cause confusion. Put another way, lines 251-252 say “... our threshold approach may prove a more sophisticated and, in the long run, more effective way to ensure sustainable grazing.” But how is this approach distinct from other explorations of grazing carrying capacities, using simulation or otherwise?

RE: Yes, we agree. Firstly, we revised the title of the paper as *“A productivity-based grazing intensity threshold as the early warning signal for grassland degradation on the Qinghai-Tibetan Plateau”* to make it clearer that the “grassland degradation” is based on the criterion of grassland net primary productivity, and the “threshold” means the grazing intensity threshold which could induce heavily grassland degradation, but not the threshold for grassland ecosystem degradation itself.

The “carrying capacity” refers to “the maximum number, density, or biomass of a population that a specific area can sustainably support” (Hartvigsen, 2022). In fact, the threshold we used in this study differs from it. We defined the threshold under the

condition that the productivity of grassland reached an extreme degradation status caused by grazing, which could lead to ecosystem collapse (NPP fell to 1% of the original NPP before application of grazing).

Correspondingly, we added the definition of “threshold” and criterion of “grassland degradation” in the introduction section following the reviewer’s suggestions as below in the revised version (see lines 107-112).

“We adopted net primary productivity (NPP) as an indicator of grassland degradation, which has proven to be a good estimator of ecosystem functioning³³ and land degradation³⁴. We assumed that grassland productivity would decrease with increasing GI, and we defined the GI threshold to be the GI at which NPP fell to 1% of the pregrazing level that was considered as an extreme degradation status for grassland (see Methods).”

We also added discussion on the issue of “carrying capacity” and estimated “threshold” in the first part of discussion section as below in the revised version (see lines 218-226).

“The GI threshold defined in this study differs from the concept of “carrying capacity” that can “sustainably support”³⁶. The GI threshold may exceed the carrying capacity because GI at the threshold level is predicted to lead to an extreme degradation which is “unsustainable”. Although many previous reports of the percentages of the entire QTP grasslands that have been degraded, the mention percentages were simply taken from earlier work that did not undergo peer review (SI02). In fact, grassland degradation could be divided into different catalogs, including lightly degraded, moderate degraded and heavily degraded grasslands. In this study, we focus on heavily degraded scenarios, which would cause ecosystem collapse.”

We rephrased the expression of lines 251-252 (in original version) and added more information as below in the revised version (see lines 281-286).

“For ensuring sustainable grazing across the entire QTP, our threshold approach based on simulations at the grid scale may prove more sophisticated and more effective in the long run, especially after further refinement, than traditional approaches based on large-scale inventories or local grazing data. Our GI threshold grids can be extended backward in time to assess whether areas have historically been overgrazed, and they can be projected into the future to estimate grazing sustainability in the face of changing climate and elevated CO₂.”

The authors fail to compare their results to the works of many others who have mapped global grassland degradation (e.g., Yang 1992; Wu et al. 2009; Ho-Young et al. 2015; Cao et al. 2019; Bardgett et al. 2021; Sun et al. 2022; Zhou et al. 2023, recognizing some are too new to be considered). Some of these works either include figures that show degradation in the QTP or are specific to the QTP. The portion of QTP degraded includes a broad range of estimates, some as high as 50%. To be clear, the authors cite

degradation of the QTP, but do not place their grazing threshold in the context of the remainder of the literature. The authors say that 80% of the QTP is below some threshold stocking level. But they also say they seek to define “the stress that grassland could take for supporting productivity before a degradation condition is triggered” lines 74-75). That suggests that the authors view at least 80% of the QTP as undegraded. Comparing the contrasting finding to that reported in the literature would improve the manuscript. Perhaps defining the threshold cited (related to the previous point) and how it relates to degradation, if it does, would be helpful.

RE: Thanks for the valuable suggestions. According to the comments, we collected references as the reviewer mentioned, as well as other relative studies.

Although many experimental studies conducted at site or local scale have evaluated the effects of grazing on grassland structure, soil qualities and other parameters related to productivity, few studies have simultaneously investigated the status of grassland degradation and grazing activities across the entire QTP. Although many previous reports of the percentages of the entire QTP grasslands that have been degraded (Cai et al., 2015; Cao et al., 2019; Dong et al., 2013; Feng et al., 2010; Liu et al., 2008; Ma & Lang, 1999; Ren et al., 2013; Sheehy, 2001; Wang et al., 2015; Wu et al., 2009; Wu et al., 2014; Zhou et al., 2023), the mention percentages were simply taken from earlier work that did not undergo peer review.

Some studies have analyzed grassland degradation on a global scale, thereby including the QTP without focusing on it. Such work may provide insights relevant for the QTP with a general information, but comparing it with our study is a little bit difficult since the authors did not report any data specifically for the Plateau (Bardgett et al., 2021; Gang et al., 2014; Gibbs & Salmon, 2015; Kwon et al., 2016).

In our study, the overgrazing status is defined as grazing intensity larger than the grazing intensity threshold. Our results showed that the overgrazing area percentage was about 20%. Considering the time until degradation across this overgrazing area, 83.0% of the area was within 20 years and 96.2% of the area was within 40 years (Figure 4e). Since we applied the grazing activities from the year of 1980, and that means the overgrazing area may have already been degraded.

In fact, grassland degradation could be divided into different catalogs, including lightly, moderate and heavily degraded grasslands. However, these catalogs may not have standard definitions. In this study, we made a definition based on NPP and focused on heavily degraded scenarios, which would cause the collapse of Tibetan grasslands. Our results figured the most serious overgrazing status based on this heavily degraded condition.

Based on the collected literatures, the rates of grassland degradation and overgrazing on the QTP or parts of the QTP were summarized in the table as below (Table R1.1, Table R1.2).

Table R1.1 Collected grassland degradation rate on the whole or parts of the QTP (NA: Not Available)

Studies	Grassland degradation declared Percentage (Based area)	Period	Indicator	Methods	Related grazing intensity information	Reference
Ma et al. (1999)	33% (QTP)	1980s~1990s	NA	NA	NA	NA
World Bank Group, (2001)	30% (Qinghai) & 26% (Tibet)	NA	NA	NA	NA	Ministry of Agriculture of China
Sheehy, (2001)	29% (Qinghai) & 18% (Tibet)	1980s	NA	NA	NA	NA
	31% (Qinghai) & 30% (Tibet)	1990s				
	21% (QTP)	1980s				
	33% (QTP)	1990s				
Berry, (2003)	30% (Qinghai) & 26% (Tibet)	NA	NA	NA	NA	Ministry of Agriculture of China
Hu and Zhang, (2003)	17% (Qinghai) & 14% (Tibet)	1990	NA	NA	NA	NA
	39% (Qinghai) & 15% (Tibet)	1999				
Wang et al., (2004)	30% (Qinghai) & 26% (Tibet)	NA	NA	NA	NA	Ministry of Agriculture of China
Lu et al. (2006)	17.0% (Qinghai) & 13.9% (Tibet)	~1990	NA	NA	NA	NA
He et al. (2008)	21.4% (QTP)	1980s	NA	NA	NA	NA
	32.7% (QTP) (1990s)	1990s				
Liu et al. (2008)	36.12% (Three-River Headwaters)	1990s	Grassland coverage gradient	Remote sensing data analysis	NA	-
Feng et al. (2010)	33% (QTP)	1980s~1990s	NA	NA	NA	Ma et al. 1999
Li et al. (2013)	33% (QTP)	1990s	NA	NA	NA	Sheehy, 2001
Ren et al. (2013)	33% (QTP)	1980s~1990s	NA	NA	NA	Feng et al. (2010)
Cai et al. (2015)*	~40% (QTP)	1990s	NA	NA	NA	Liu et al. (2008)
Wang et al. (2016)	38.8% (QTP)	2000s	NPP	remote sensing data based model (CASA)	NA	-
Zhou et al. (2023)**	32.69% (QTP)	1990s	NA	NA	NA	Foggin et al. 1996

*note: Actually, the study of Liu et al. (2008) was conducted only over Three-River Headwaters based on remote sensing data, and found that the 36.12% of grassland in this area was degraded between 1990s and 2004.

**note: Did not find the specific percentage number in the study of Foggin et al. (1996)

Table R1.2 Collected grassland overgrazing rate on the whole or parts of the QTP (NA: Not Available)

Studies	Overgrazing rate declared Percentage (Based area)	Period	Methods	Reference
Yang and Yang 2000	28.6% (QTP)	1990s	Based on grassland potential productivity calculated using Miami model	-
Qian et al. 2007	16% (Qinghai) & 78% (Tibet)	2003~2004	Based on grassland potential productivity calculated based on NDVI data	-
Zhang et al. 2014	27~89% (Different regions across QTP)	1996~2008	NA	Yang and Yang 2000 Qian et al. 2007
Supervision report of national grasslands in 2006	39% (Qinghai) & 38% (Tibet)	2006	NA	Ministry of Agriculture of People's Republic of China (Report could be obtained from http://www.moa.gov.cn/ or http://www.forestdata.cn/index.html)
Supervision report of national grasslands in 2007	38% (Qinghai) & 40% (Tibet)	2007	NA	
Supervision report of national grasslands in 2008	37% (Qinghai) & 38% (Tibet)	2008	NA	
Supervision report of national grasslands in 2009	26% (Qinghai) & 39% (Tibet)	2009	NA	
Supervision report of national grasslands in 2010	25% (Qinghai) & 38% (Tibet)	2010	NA	
Supervision report of national grasslands in 2011	25% (Qinghai) & 32% (Tibet)	2011	NA	
Supervision report of national grasslands in 2012	16% (Qinghai) & 29% (Tibet)	2012	NA	
Supervision report of national grasslands in 2013	14% (Qinghai) & 22% (Tibet)	2013	NA	
Supervision report of national grasslands in 2014	13% (Qinghai) & 19% (Tibet)	2014	NA	
Supervision report of national grasslands in 2015	13% (Qinghai) & 19% (Tibet)	2015	NA	
Supervision report of national grasslands in 2016	11.9% (Qinghai) & 16% (Tibet)	2016	NA	
Supervision report of national grasslands in 2017	9.8% (Qinghai) & 13% (Tibet)	2017	NA	

According to the reviewer's suggestion, we made a comparison between our results and that showed in other collected studies in the discussion section as below in the revised version (see lines 227-232).

"Our simulations suggest that overgrazing and degradation affect around 20% of QTP grasslands, based on ratios of GI to the threshold. This figure is comparable to the multi-year mean grassland overgrazing rates of 16% for Qinghai and 24% for Tibet

between 2010 and 2017, though several studies have suggested degradation rates of 21-40% across the entire QTP from the 1980s to 2000s (Supplementary Information 2). Our results figured the most serious overgrazing status based on heavily degraded scenarios.”

Based on the collected literatures, we also added a short review (including two tables above) on the degradation and overgrazing status of grassland on the QTP as an additional SI section in the revised version (see SUPPLEMENTARY INFORMATION 2 (SI02)).

The authors then calculate (lines 137-143 and many other locations) how long areas with grazing exceeding their threshold would take to become degraded. That suggests that the authors see only some small fraction of the 20% of land exceeding the threshold as currently degraded. Contradictions in this regard in the writing itself and in the writing compared to the literature exist and should be rectified.

RE: Thanks for pointing out this. For those overgrazing area (~20% in our study), model results indicate the grassland would be degraded within 20 years (83.0% of the overgrazing area) or within 40 years (96.2% of the overgrazing area) if the grazing activities applied at an actual level (Figure 4e). In our study, we applied the grazing activities from the year of 1980, and that means those areas of overgrazing may have already been degraded. According to the reviewer’s comments, we added this information in the revised version as below to avoid confusion (see lines 159-162).

“Modeling of these areas indicated that 96.2% of the overgrazed grassland would be degraded within 40 years or 83.0% within 20 years if GI remained at current levels (Figure 4e). Since we applied the grazing activities in the simulation from 1980 onwards, it indicated that the grasslands in these areas would be degraded by now.”

I invite the authors to consider the section starting at line 158 for review. It is not as clear as the earlier material. I think it is because the actual grazing intensity and threshold intensities inform statistics, and so two percentages are often provided, which is a little confusing. So “... nearly 89% and 72% of the GI ratio larger than 0.5 and 0.6 under normal grazing condition, respectively” and “... to offset the negative effects of grazing with area percentage of 72% or 89% ...” is a little tough going for readers. The same information may be shared in almost as pithy a way while untangling the combined statements.

RE: Yes, we rephrased this section and revised the corresponding Figure 5. We cited the percentage numbers as showed exactly in the figure into the main text to avoid confusion as below in the revised version. (see lines 171-196).

We made an additional simulation that the grazing intensity was changed to make ratio of GI to threshold was reduced to 0.6 in areas where the ratio was > 0.6. The results were also added to Figure 5 to show a suitable ratio of GI to threshold (60%, ranged from 50% to 70%) may maintain the balance between grassland protection and human

demands. (also see lines 171-196).

“Simulations suggested that climate change and elevated atmospheric CO₂ concentration would partly offset the negative effects of grazing activities on 68.3% of QTP grasslands (Figure 5a, b and Supplementary Information 1), nearly all of which (98.5%) was subject to normal grazing and which accounted for 83.6% of the total grassland area subject to normal grazing (Figure 5b). Across these “offset” areas, 69.2% showed a ratio of GI to the threshold below 0.6, while 83.0% showed a ratio below 0.7 (Figure 5c). The beneficial offsets were observed in only 5.3% of overgrazed areas (Figure 5b), 76.9% of which featured ratios of GI to the threshold below 3.0 (Figure 5d). Among the remaining 31.7% of QTP grassland areas where climate change and elevated CO₂ were not predicted to offset negative effects of grazing (Figure 5b), 58.5% were overgrazed (Figure 5b), with ratios of GI to threshold from 2.0 to 6.0 (Figure 5e), and 41.5% were subject to normal grazing (Figure 5b), of which 71.3% showed a ratio of GI to threshold larger than 0.6 and 88.4% showed a ratio of GI to threshold larger than 0.5 (Figure 5f). Although the areas in “non-offset areas” under normal grazing, the positive effects of climate change and elevated CO₂ could not compensate the negative effects of grazing on grassland productivity.

Our model suggested that setting the ratio of GI to threshold between 0.5 and 0.7 would preserve 70-80% of the areas that were currently subject to normal grazing and were predicted to experience positive offsets from climate change and CO₂ (Figure 5c, g). This range from 0.5 to 0.7 showed the greatest potential for maximizing the areas subject to normal grazing or overgrazing (both currently in “non-offset areas”) that could benefit from positive offsets due to climate change or CO₂ (Figure 5b, e, f). When we reduced the ratio of GI to threshold to 0.6 in areas where the ratio was > 0.6 in an additional simulation, we found that the positive effects of climate change and elevated CO₂ could offset the negative effects of grazing on over 89.1% of QTP grassland areas (Figure 5 h), by “converting” 70% of currently “non-offset areas” into “offset areas” (Figure 5b, h). The “converted” areas come from 78% of overgrazing area and 58% of normal grazing area (Figure 5b, h).”

The revised Figure 5 showed as below (Figure R1.1):

Figure R1.1 (also Figure 5) Ratios of actual GI to the threshold on QTP grasslands. a. Distribution of areas where the negative effects of grazing are predicted to be offset by the positive effects of climate change and elevated CO₂ on grassland NPP (“offset areas”) and areas where such offset is not predicted (“non-offset areas”); b. The bar charts indicated the area percentage of “offset areas” and “non-offset areas” of the grassland on QTP, along with the area percentage (white numbers) of overgrazing (OG, red color) and normal grazing (NG, green color) over each region, as well as the area percentage of OG (red color; blue numbers) and NG (green color; yellow numbers) distributed in offset areas and non-offset areas respectively; c-d. The area percentage histogram and area percentage cumulative curve of the ratio of GI to GI threshold for the “offset areas” subject to normal grazing (NG, green color) (c) or overgrazing (OG, red color) (d); e-f. The area percentage histogram and area percentage cumulative curve of the ratio of GI to GI threshold for the “non-offset areas” of OG (red color) (e) or NG (green color) (f), and different from others, the area percentage of GI ratio in panel f accumulated decreasingly from 0.95 to 0.15; g. The merged area percentage cumulative curves of the ratio of GI to GI threshold subject to NG (green color), extracted from panel c (blue line) and f (red line). h. The bar charts indicate the area percentage of “offset areas” and “non-offset areas” of QTP grasslands, along with the area percentage (white numbers) of OG (red color) or NG (green color) over each region based on an additional simulation that was performed in which the ratio of GI to threshold was set to be 0.6 where the ratio was > 0.6.

Some observed grazing rates converted to categories were used in assessment, which is helpful. It seems there remains a great deal of remotely sensed information amenable to these broad-scale analyses that is suitable for simulation assessment as well, and readily available. Typical data used for a model that has NPP estimates at its core would

be NPP and GPP estimates, plus NDVI or EVI, from MODIS, for example. The authors may wish to describe why those data were not used in assessment or incorporate their use.

RE: According to reviewer's suggestion, multi-year average NPP (2000-2015) simulated in this study was compared with the NPP products retrieved from remote sensed data of AVHRR and MODIS at grid scale across the QTP. The results and figures were added in the model performance evaluation section as below in the revised version (see lines 529-536) and the supplementary materials (see Figure S3.9).

“Multi-year average NPP (2000-2015) was compared between simulated in this study and the productions retrieved from remote sensed data of AVHRR and MODIS at grid scale across QTP (Figure S3.9) for a reference of model performance evaluation. The simulated NPP values agreed better with the AVHRR data ($R^2 = 0.62$, Figure S3.9b) than with the MODIS data ($R^2 = 0.47$, Figure S3.9a). Simulated NPP values showed a coefficient of determination of nearly 0.6 when compared with NPP values averaged between the MODIS and AVHRR datasets (Figure S3.9c). The AVHRR data usually contained greater NPP values than the MODIS data (Figure S3.9d).”

Line 33. There is ambiguity in the 70% figure. Is it 70% of the area in question? Or 70% of the effect offset? Something else?

Re: It is 70% of the total grassland area of the QTP. We rephrased the abstract in the revised version. (see lines 27-30).

“.....The positive effects of climate change and elevated CO₂ can partly offset the negative effects of grazing across nearly 70% of grassland on the Plateau, but only in areas where current GI does not exceed the GI threshold.”

Line 48. Ensuring the “ecological security of the QTP” may be briefly expanded upon. I'm curious, anyway.

Re: Yes, to make it clearer, we modified the expression as “*Such alpine grasslands provide forage for 12 million yak and 30 million sheep and goats, and their responses and resilience to climate change and human activities strongly affect the livelihood of about 5 million pastoralists and agropastoralists on the QTP*”. (also see lines 42-45).

The original “ecological security” means the resilience of grassland ecosystem, referred from *Ecological Security: A Definition. (2021). In M. McDonald (Ed.), Ecological Security: Climate Change and the Construction of Security (pp. 94-122). Cambridge: Cambridge University Press (McDonald, 2021).*

Line 102. “Processed model” sounds a little unusual. Perhaps “process-based model”?

Re: Yes, it's revised as suggested.

Line 112. Briefly define sheep units for those working in other regions.

Re: Yes, it's revised as suggested.

Lines 163-164. “To some extent” may be removed given that “partly” was used.

Re: Yes, it’s revised as suggested.

Line 167. GI is introduced without having been defined.

Re: Yes, we checked through the MS and defined GI at its first appearance.

Lines 251-252 introduce a comparison without the thing being compared against. It speaks of a more effective way, compared to ... I’m not speaking of rangeland science here, only the writing. “compared to traditional approaches” would be sufficient, for example.

Re: Yes, lines 251-252 were revised as mentioned before (also see lines 281-286).

May lines 413-415 (“When the NPP of a cell reached nearly to zero under a certain grazing intensity, then we defined that grazing intensity to be the threshold value for the grid cell, such that intensities above that value would lead to degradation.”) somehow find their way into the body of the manuscript? This is really the heart of the matter, the means in which you defined a threshold value (the confusing name aside). A summary of this approach in the Introduction or Results would be very helpful, and require just a few words (e.g., a threshold was crossed when NPP was < 0 [but using another word for “threshold”]).

Re: Yes, according to the reviewer’s suggestion, we defined the grassland degradation and grazing intensity threshold criterion in the introduction section as mentioned above (also see lines 107-112).

Through the whole text, we made clearly that the “threshold” means “grazing intensity threshold”. It could induce heavily grassland degradation based on productivity, leading to the collapse of grasslands. It is not the threshold for grassland ecosystem degradation itself.

Reviewer #2 (Remarks to the Author):

The manuscript entitled “Early warning of grazing-induced thresholds of grassland degradation on the Qinghai-Tibetan Plateau”, by Zhu et al. addresses the impact of grazing intensity on net primary productivity (NPP) along the Qinghai-Tibetan Plateau (QTP) grassland ecosystems, using a process-based ecosystem modeling. The particular objectives were (i) to determine the grazing intensities that induce an abrupt reduction in NPP, (ii) to apply these findings to identify grassland areas under threat of degradation, and (iii) to predict when such degradation might occur under current and future grazing and climate conditions. The main results indicate that most grasslands of the QTP are managed with grazing intensities below those that induce the abrupt reduction in NPP, and that maintaining the grazing intensity around 60% of that induces the abrupt change has the potential to sustain the human demand without inducing grassland degradation. These aspects are critical since this extensive area is mostly covered by grasslands that sustain millions of domestic herbivores, which in turn are the livelihoods of millions of people. The manuscript is clear and the study is well performed. The findings are novel and have the potential to generate some international interest in its field of study. However, there are some issues that should be addressed.

RE: Thanks for the reviewer’s positive feedbacks and constructive comments! Our responses are listed below point by point.

My main concern is the unique ecological attribute and the relationship to estimating thresholds (grazing intensity and NPP), which can lead to misunderstanding degradation processes since the study did not consider other critical ecological attributes and other aspects of grazing management beyond the stocking rate (here grazing intensity). On the one hand, degradation processes can be evidenced through ecosystem attributes different from NPP, which can be more appropriate to early detect degradation (e.g. Kéfi et al., 2007; Berdugo et al., 2017, 2020; Oñatibia & Aguiar, 2023). For example, degradation may occur without inducing changes in NPP (see Verón et al., 2006). On the other hand, grazing management strategies are much more complex than grazing intensity. For example, grazing and rest periods and the spatial distribution of herbivores in the landscape are key factors to control land degradation and promote productivity (see e.g. Briske, 2017). I suggest the authors expand the section on potential limitations of the study including a discussion of these issues.

RE: We totally agree with the reviewer that the criteria for grassland degradation should be evaluated comprehensively based on different indicators related to plant and soil of grassland ecosystem, including composition, structure, and function of the grassland ecosystem. In this study, we took NPP as the indicator of grassland degradation as this is an integrated and principal indicator, especially for regional evaluations. We also agree that grazing management strategies are much more complex than grazing intensity.

According to the reviewer’s suggestion and the provided references, we expanded the discussion on potential limitation of our study in the discussion section related to

above issues as below in the revised version. (see lines 309-311 and lines 277-279).

“Refinement of our method of calculating the GI threshold for grassland degradation will require going beyond NPP to take into account multiple indicators related to the plants and soil of the grassland ecosystem^{10, 11, 41, 46, 47, 48}.”

“Although we focused on GI in this study, grazing management strategies need to consider additional factors affecting land degradation and productivity, such as the time of grazing and resting and the spatial distribution of herbivores⁴⁰.”

Besides, both contextualizing the study and comparing the main findings with existing global literature (other than just for the QTP) on early warning signals of degradation and ecosystem thresholds are also recommended.

RE: Yes, we agree. Although many experimental studies conducted at site or local scale have evaluated the effects of grazing on grassland structure, soil qualities and other parameters related to productivity, few studies have simultaneously investigated the status of grassland degradation and grazing activities across the entire QTP. Although many previous reports of the percentages of the entire QTP grasslands that have been degraded (Cai et al., 2015; Cao et al., 2019; Dong et al., 2013; Feng et al., 2010; Liu et al., 2008; Ma et al., 1999; Ren et al., 2013; Sheehy, 2001; Wang, et al., 2015; Wu et al., 2009; Wu et al., 2014; Zhou et al., 2023), the mention percentages were simply taken from earlier work that did not undergo peer review.

Some studies have analyzed grassland degradation on a global scale, thereby including the QTP without focusing on it. Such work may provide insights relevant for the QTP with a general information, but comparing it with our study is a little bit difficult since the authors did not report any data specifically for the Plateau (Bardgett et al., 2021; Gang et al., 2014; Gibbs et al., 2015; Kwon et al., 2016).

In our study, the overgrazing status is defined as grazing intensity larger than the grazing intensity threshold. Our results showed that the overgrazing area percentage was about 20%. Considering the time until degradation across this overgrazing area, 83.0% of the area was within 20 years and 96.2% of the area was within 40 years (Figure 4e). Since we applied the grazing activities from year of 1980, and that means the overgrazing area may have already been degraded.

In fact, grassland degradation could be divided into different catalogs, including lightly degraded, moderate degraded and heavily degraded grasslands. However, these catalogs may not have standard definitions. In this study, we made a definition based on NPP and focused on heavily degraded scenarios, which would cause ecosystem collapse. Our results figured the most serious overgrazing status based on this heavily degraded condition.

Based on the collected references, the rates of grassland degradation and overgrazing on the QTP or parts of the QTP were summarized in the table as below (Table R2.1, Table R2.2).

Table R2.1 Collected grassland degradation rate on the whole or parts of the QTP (NA: Not Available)

Studies	Grassland degradation declared Percentage (Based area)	Period	Indicator	Methods	Related grazing intensity information	Reference
Ma et al. (1999)	33% (QTP)	1980s~1990s	NA	NA	NA	NA
World Bank Group, (2001)	30% (Qinghai) & 26% (Tibet)	NA	NA	NA	NA	Ministry of Agriculture of China
Sheehy, (2001)	29% (Qinghai) & 18% (Tibet)	1980s	NA	NA	NA	NA
	31% (Qinghai) & 30% (Tibet)	1990s				
	21% (QTP)	1980s				
	33% (QTP)	1990s				
Berry, (2003)	30% (Qinghai) & 26% (Tibet)	NA	NA	NA	NA	Ministry of Agriculture of China
Hu and Zhang, (2003)	17% (Qinghai) & 14% (Tibet)	1990	NA	NA	NA	NA
	39% (Qinghai) & 15% (Tibet)	1999				
Wang et al., (2004)	30% (Qinghai) & 26% (Tibet)	NA	NA	NA	NA	Ministry of Agriculture of China
Lu et al. (2006)	17.0% (Qinghai) & 13.9% (Tibet)	~1990	NA	NA	NA	NA
He et al. (2008)	21.4% (QTP)	1980s	NA	NA	NA	NA
	32.7% (QTP) (1990s)	1990s				
Liu et al. (2008)	36.12% (Three-River Headwaters)	1990s	Grassland coverage gradient	Remote sensing data analysis	NA	-
Feng et al. (2010)	33% (QTP)	1980s~1990s	NA	NA	NA	Ma et al. 1999
Li et al. (2013)	33% (QTP)	1990s	NA	NA	NA	Sheehy, 2001
Ren et al. (2013)	33% (QTP)	1980s~1990s	NA	NA	NA	Feng et al. (2010)
Cai et al. (2015)*	~40% (QTP)	1990s	NA	NA	NA	Liu et al. (2008)
Wang et al. (2016)	38.8% (QTP)	2000s	NPP	remote sensing data based model (CASA)	NA	-
Zhou et al. (2023)**	32.69% (QTP)	1990s	NA	NA	NA	Foggin et al. 1996

*note: Actually, the study of Liu et al. (2008) was conducted only over Three-River Headwaters based on remote sensing data, and found that the 36.12% of grassland in this area was degraded between 1990s and 2004.

**note: Did not find the specific percentage number in the study of Foggin et al. (1996)

Table R2.2 Collected grassland overgrazing rate on the whole or parts of the QTP (NA: Not Available)

Studies	Overgrazing rate declared Percentage (Based area)	Period	Methods	Reference
Yang and Yang 2000	28.6% (QTP)	1990s	Based on grassland potential productivity calculated using Miami model	-
Qian et al. 2007	16% (Qinghai) & 78% (Tibet)	2003~2004	Based on grassland potential productivity calculated based on NDVI data	-
Zhang et al. 2014	27~89% (Different regions across QTP)	1996~2008	NA	Yang and Yang 2000 Qian et al. 2007
Supervision report of national grasslands in 2006	39% (Qinghai) & 38% (Tibet)	2006	NA	Ministry of Agriculture of People's Republic of China (Report could be obtained from http://www.moa.gov.cn/ or http://www.forestdata.cn/index.html)
Supervision report of national grasslands in 2007	38% (Qinghai) & 40% (Tibet)	2007	NA	
Supervision report of national grasslands in 2008	37% (Qinghai) & 38% (Tibet)	2008	NA	
Supervision report of national grasslands in 2009	26% (Qinghai) & 39% (Tibet)	2009	NA	
Supervision report of national grasslands in 2010	25% (Qinghai) & 38% (Tibet)	2010	NA	
Supervision report of national grasslands in 2011	25% (Qinghai) & 32% (Tibet)	2011	NA	
Supervision report of national grasslands in 2012	16% (Qinghai) & 29% (Tibet)	2012	NA	
Supervision report of national grasslands in 2013	14% (Qinghai) & 22% (Tibet)	2013	NA	
Supervision report of national grasslands in 2014	13% (Qinghai) & 19% (Tibet)	2014	NA	
Supervision report of national grasslands in 2015	13% (Qinghai) & 19% (Tibet)	2015	NA	
Supervision report of national grasslands in 2016	11.9% (Qinghai) & 16% (Tibet)	2016	NA	
Supervision report of national grasslands in 2017	9.8% (Qinghai) & 13% (Tibet)	2017	NA	

According to the reviewer's suggestion, we made a comparison between our results and that showed in other collected studies in the discussion section as below in the revised version (see lines 227-232).

"Our simulations suggest that overgrazing and degradation affect around 20% of QTP grasslands, based on ratios of GI to the threshold. This figure is comparable to the multi-year mean grassland overgrazing rates of 16% for Qinghai and 24% for Tibet

between 2010 and 2017, though several studies have suggested degradation rates of 21-40% across the entire QTP from the 1980s to 2000s (Supplementary Information 2). Our results figured the most serious overgrazing status based on heavily degraded scenarios.”

Based on the collected literatures, we also added a short review (including two tables above) on the degradation and overgrazing status of grassland on the QTP as an additional SI section in the revised version (see SUPPLEMENTARY INFORMATION 2 (SI02)).

Also, in the revised version, we modified the title as “A productivity-based grazing intensity threshold as the early warning signal for grassland degradation on the Qinghai-Tibetan Plateau” to avoid confusion. In this study, “threshold” means the “grazing intensity threshold” that induces grassland heavy degradation, but not the threshold for grassland ecosystem degradation itself. We think the early warning is specified to the grazing intensity threshold, based on e.g. “time until degradation”, and we hope the study can provide an early implication for grazing activities management on the QTP and preventing the grassland from degradation. (as below or see lines 275-277).

“From an early warning signal of the grazing intensity threshold determined by “time until degradation”, an early implication was supposed to provide for grazing activities management on the QTP and preventing the grassland from degradation.”

Regarding the model validation, the field sites are relatively few and cover a small area compared to the large extent of the whole study area. This weakness is not fatal, but the authors should discuss this aspect, maybe also expanding the section about the potential limitations of the study.

RE: Yes, according to the reviewer’s comments, we expanded the discussion about the limited covering data on model performance evaluation as below in the revised version. (see lines 326-327).

“Finally, the model performance evaluation should be improved by including more field data covering the entire QTP.”

Also, we added a comparison of multi-year average NPP between that simulated in this study and retrieved from remote sensed data of AVHRR and MODIS at grid scale across QTP as below in the revised version (see lines 529-536, Figure S3.9).

“Multi-year average NPP (2000-2015) was compared between simulated in this study and the productions retrieved from remote sensed data of AVHRR and MODIS at grid scale across QTP (Supplementary Fig. S3.9) for a reference of model performance evaluation. The simulated NPP values agreed better with the AVHRR data ($R^2 = 0.62$, Supplementary Fig. S3.9b) than with the MODIS data ($R^2 = 0.47$, Supplementary Fig. S3.9a). Simulated NPP values showed a coefficient of determination of nearly 0.6 when compared with NPP values averaged between the MODIS and AVHRR datasets

(Supplementary Fig. S3.9c). The AVHRR data usually contained greater NPP values than the MODIS data (Supplementary Fig. S3.9d)."

Finally, a minor issue is that the treatment given to the term "threshold" is vague and sometimes unclear (e.g. lines 67, 74, 78, 88, 102, and see specific comments below). This is important since a central role is given to this term throughout the manuscript.

RE: Yes, in the revised version, we clearly defined grassland degradation and grazing intensity threshold criterion adopted in our study in the introduction section as below (see lines 107-112).

"We adopted net primary productivity (NPP) as an indicator of grassland degradation, which has proven to be a good estimator of ecosystem functioning³³ and land degradation³⁴. We assumed that grassland productivity would decrease with increasing GI, and we defined the GI threshold to be the GI at which NPP fell to 1% of the pregrazing level that was considered as an extreme degradation status for grassland (see Methods)."

Through the whole text, we made clearly that the "threshold" means "grazing intensity threshold" that could induce grassland degradation based on productivity, but not the threshold for grassland ecosystem degradation itself.

For the line 67 (in original version), we revised the sentence as *"Even slight changes in climate and human factors can substantially affect the structure and functioning of alpine grasslands on the QTP^{14, 15, 16}, leading to sudden, potentially irreversible changes in ecology and functioning^{17, 18}".* (see lines 63-65).

For the line 74 (in original version), we revised the sentence as *"Given the pertinence of grazing to QTP grasslands^{8, 19}, we propose to develop a "grazing intensity (GI) threshold" which would induce an extreme grassland degradation to provide an early implication for grazing activities management."* (see lines 70-72).

For the line 78 (in original version), we revised the sentence as *"...based on a predicted GI threshold that could induce grassland degradation may support..."* (see lines 74-75).

For the line 88 (in original version), we revised the sentence as *"...in depth understanding of a GI threshold that could induce grassland degradation at regional scale across the QTP..."* (see lines 86-87).

For the line 102 (in original version), we revised the sentence as *"...a grazing framework for grassland ecosystems was modified and integrated into a process-based model to explore a GI threshold inducing grassland degradation on the QTP..."* (see lines 105-107).

In general, it is not completely clear how it relates to an early warning signal for grassland degradation. Definitions of threshold and degradation (and their relationship) from the introduction will help to understand the main ideas addressed. A more precise alternative could be directly referring to grazing-induced abrupt NPP reduction instead of threshold and degradation.

RE: Yes, according to the reviewer’s comments, we clearly defined the criterion of grassland degradation and grazing intensity threshold adopted in our study in the introduction section as below in the revised version. (see lines 107-112).

“We adopted net primary productivity (NPP) as an indicator of grassland degradation, which has proven to be a good estimator of ecosystem functioning³³ and land degradation³⁴. We assumed that grassland productivity would decrease with increasing GI, and we defined the GI threshold to be the GI at which NPP fell to 1% of the pregrazing level that was considered as an extreme degradation status for grassland (see Methods).”

Also, we revised the title as *“A productivity-based grazing intensity threshold as the early warning signal for grassland degradation on the Qinghai-Tibetan Plateau”* to avoid confusion. In this study, “threshold” means the “grazing intensity threshold”, but not the threshold for grassland ecosystem degradation itself. We think the early warning is specified to the grazing intensity threshold, based on e.g. “time until degradation” (Figure R2.1), and we hope the study can provide an early implication for grazing activities management on the QTP and preventing the grassland from degradation.

Figure R2.1 (also Figure S3.5) Schematic of how changes in net primary productivity (NPP) of grasslands subject to different grazing intensities (GIs) can be used to determine “grazing intensity (GI) threshold” and “time until degradation”. The process assumed that grassland productivity would decrease as GI increased. The model was run under different GIs, from non-grazing (GI =

0.0 SU ha⁻¹ year⁻¹) to maximum possible grazing (GI = 15 SU ha⁻¹ year⁻¹), increasing in a step of 0.5 SU ha⁻¹ year⁻¹ for a period of 120 years. When NPP fell to 1% of the original NPP before application of the GI, that GI was defined as the GI threshold. The interval between onset of GI and when NPP fell to 1% of the original NPP was defined as the time until degradation.

SPECIFIC COMMENTS

Lines 1-2. I suggest revisiting the title after considering the comments on what thresholds represent and how they are related to early warning signals of grassland degradation.

RE: Yes, according to the comment, we revised the title as “A productivity-based grazing intensity threshold as the early warning signal for grassland degradation on the Qinghai-Tibetan Plateau” to avoid confusion. We think the early warning is specified to the grazing intensity threshold, based on e.g. “time until degradation”, and we hope the study can give an early implication for grazing activities management on the QTP and preventing the grassland from degradation.

Lines 26-27. Suitable threshold? Threshold of what? Shouldn't it be “suitable grazing intensity”?

RE: Yes, we rephrased the abstract in the revised version. (see lines 23-26).

“Based on an improved process-based ecosystem model, the present work defined a productivity-based grazing intensity (GI) threshold that induces grassland extreme degradation to assess whether and where current grazing activity is sustainable.”

Lines 50-51. The authors indicate that QTP grasslands are very fragile. Please, justify this statement and add citations. If this information is that found in lines 64-66, please, avoid repetition.

RE: Yes, we added references for this statement. And deleted the redundant information in the text.

Lines 58-62. Please define degradation. A definition of what grassland degradation represents to the authors would help make this section clearer.

RE: Yes, we removed some redundant information and rephrased this part as below in the revised version. (see lines 51-56).

“Earlier in history, animal farmers tended to maintain relatively small herds just large enough to support their nomadic lifestyle, with some extra animals to buffer the effects of harsh weather. The grazing of these small herds was sustainable because it did not lead to irreversible grassland degradation, defined as a decline in the quality of plants and soils, as well as changes in ecosystem composition, structure, and function 10, 11.”

We also added definition of grassland degradation used in this study in the final part of the introduction section as mentioned above. (also see lines 107-112).

Line 140. Remove “that”.

RE: Yes, it’s revised as suggested.

Lines 220-227. Areas described as overgrazed seem to be generally the most arid (e.g. northwest). If this is correct, authors can discuss the potential interaction between increasing grazing and aridity (see e.g. Gaitán et al., 2018; Oñatibia et al., 2020).

RE: Thanks for the suggestion. We agree that aridity (or drought trend) and grazing pressure may have a negative synergistic effect on dryland ecosystem as mentioned in the study of Gaitán et al. (2018) and Oñatibia et al. (2020). For the grassland ecosystem on QTP in this study, we found a warming and humidification trend (Figure S2.1). We also checked the multi-year mean precipitation of the periods of 2000-2009 and 2010-2017 and the maps showed below (Figure R2.2a, Figure R2.2b). The precipitation in northwest area of the QTP mostly ranged from 300 to 400 mm/year and the area with precipitation in this range is expanding (Figure R2.2a, Figure R2.2b). A Palmer Drought Severity Index (PDSI) map for the QTP was extracted from a global PDSI data set created by van der Schrier et al. (2013), which was downloaded from <ftp://ftp.ceda.ac.uk> (Figure R2.2c). Based on the classification of the PDSI (Table R2.3), we can see most area of the northwest area of the QTP is not in dry condition. Therefore, the aridity may not be enhanced for the northwestern part during the studying period.

Figure R2.2 a. Multi-year mean precipitation on QTP for the period of 2000-2009; b. Multi-year mean precipitation on QTP for the period of 2010-2017; c. PDSI map of QTP extracted from data set of PDSI_CRU_TS_3.10.01_based between 2001 and 2009, <ftp://ftp.ceda.ac.uk>

Table R2.3 PDSI classification in the study of van der Schrier et al. (2013)

Table 1. Classification of Dry and Wet Conditions as Defined by Palmer [1965] for the PDSI

PDSI	Class
≥ 4.0	extremely wet
3.0 : 4.0	severely wet
2.0 : 3.0	moderately wet
1.0 : 2.0	slightly wet
0.5 : 1.0	incipient wet spell
-0.5 : 0.5	near normal
-0.5 : -1.0	incipient dry spell
-1.0 : -2.0	slightly dry
-2.0 : -3.0	moderately dry
-3.0 : -4.0	severely dry
≤ -4.0	extremely dry

Lines 366-368. If grazing intensity is the annual average, its units could be SU/ha/year (stocking rate units) instead of density units (SU/ha).

RE: Yes, we revised the unit as suggested SU/ha/year all through the text, figures and tables.

Line 425. I suggest moving this Figure with an expanded legend to the main manuscript.

RE: Yes, the grazing intensity threshold map under future climate change was presented in the main text as Figure 6.

Line 608-609. I suggest that authors indicate what the white color represents on the map; if it means the absence of data or if represents other vegetation types different from steppes and meadows.

RE: The white color on the map represents other land cover types different from steppes and meadows. We added this information in all the related Figure legends in the paper as suggested.

Figure legends. Information is generally lacking for the figures to be self-explanatory, particularly those in the supplementary material.

RE: Yes, we added more information on figure legends or on the figures themselves in the revised version.

Reviewer #3 (Remarks to the Author):

The manuscript “Early warning of grazing-induced thresholds of grassland degradation on the Qinghai-Tibetan Plateau” by Zhu & Chen et al. attempts to identify grazing intensity thresholds that would lead to grassland degradation. The question is of broad relevance, and such results could potentially be used as the basis for future grazing management in the area.

RE: Thanks for the reviewer’s positive feedbacks and comments and the responses are listed below point by point.

Overall the manuscript did not provide enough information for me to assess the results and conclusions drawn critically. The model, in particular the vegetation dynamical component, is not sufficiently described in either the main text or supplementary materials to allow appropriate assessment of the presented results and conclusions. The authors refer to two published manuscripts (Zhang et al., 2017; Zhu et al., 2014). I have endeavoured to follow the cited manuscripts to gain a better understanding of the model used but have done so unsuccessfully. In Zhu et al., (2014), the model description leads to a string of further citations: “The TRIPLEX- GHG model (Peng et al., 2013) is based on the legacy of well-established and published models that include IBIS (Foley et al., 1996), DNDC (Li, 2000), TRIPLEX (Peng et al., 2002) and CASACNP (Wang et al., 2010). ”. The cited Peng et al. (2013) paper does not appear to exist. At best, I can find an abstract submitted to the European geoscience Union (EGU), which is associated with TRIPLEX-GHG. This manuscript, however, indicates that IBIS represents the vegetation model component. Zhang et al. (2017) explicitly cite the above-mentioned EGU abstract (Integrating greenhouse gas emission processes into a dynamic global vegetation model of TRIPLEX-GHG) at the beginning of the description of the model. Model improvements by (Liu et al., 2005) are also mentioned here. Based on the equation for NPP provided by the authors (line 303), I am guessing that the model version they are using is not that developed by Liu et al. (2005) and does not include any nutrient constraints on growth or photosynthesis. This is guesswork, though, and I did not see any code submitted, which could have been checked.

RE: Thanks to the reviewer for the comments. We apologized for the lack of information on the vegetation dynamical component. The vegetation dynamical module, land surface module, plant phenology module, and soil biogeochemical module of TRIPLEX-GHG is primary based on the model of IBIS (Foley et al., 1996; Kucharik et al., 2000). We declared this in our previous paper of Zhu et al. (2014) and Zhang et al. (2017). And currently we have added different modules into the model including greenhouse gases emission (CH₄/N₂O) process, nitrogen and phosphorus cycling process, wetland area dynamic process, peatland carbon accumulation process, and grassland grazing process, *etc.* The two citations of the model (Peng et al., 2013 on North America Carbon Program and Peng et al. 2013 on EGU) mentioned by the reviewer were presented on two meetings with the purpose of model prototype introduction which were not peer reviewed. The paper of Zhu et al. (2014) is the first

publication for the model of TRIPLEX-GHG. Model improvements by Liu et al. (2005) (who is the fourth author of this paper) are fully coupled in current version (also referred in Zhu et al. 2014 and Zhang et al. 2017) and applied in this study. So, nitrogen constraints on NPP were considered in this study. In the revised version, we added more information about the model description in the methodology section as below in the revised version (see lines 332-347).

“The TRIPLEX-GHG model is a dynamic global vegetation model that takes into account land surface processes, vegetation phenology, long-term vegetation dynamics, soil biogeochemical processes, greenhouse gas emissions, cycling of nitrogen and phosphorus, wetland area dynamics, and peatland carbon accumulation^{56, 57, 58, 59}. The TRIPLEX-GHG modules of vegetation dynamics, land surface, plant phenology, and soil biogeochemistry are based primarily on the Integrated Biosphere Simulator (IBIS) model^{60, 61} (Supplementary Fig. S3.3). The vegetation in TRIPLEX-GHG was represented by plant functional types (PFTs) and two types of grasses (C3 and C4) were included. Vegetation dynamics was characterized by PFT changes in terms of biomass and leaf area index, with competition among PFTs for sunlight and water. Plant phenology was modeled based on the coldest temperatures, thresholds for accumulated growing degree days, and thresholds for 10-day average temperature. Canopy photosynthesis was modeled based on the Farquhar model to calculate gross primary productivity (GPP) and net primary productivity (NPP) for each PFT^{60, 61}. An improvement considering nitrogen constraints on estimation of canopy NPP was adopted by applying a soil N availability modifier and dynamic C:N ratios in order to capture how the carbon cycle responds to dynamic nitrogen availability⁵⁸.”

And, we provided the source code with example input data in the manuscript submission system for the reviewers' information.

One of the major results presented by Zhu & Chen et al. is that the positive effects of climate change can offset, to a large extent, grazing. Based on the evidence presented, it is not clear whether such positive effects are as a result of temperature, precipitation, or increasing CO₂. Presumably (this is not stated), the authors have turned off all other plant functional types other than temperate C3 grasses or similar for the present study. The extent to which CO₂ may fertilise C3 grasses is uncertain (Reich et al., 2018). It is not clear what IBIS version the authors used and whether this version limits photosynthetic rate via any mechanism, e.g. nutrient limitation.

RE: Good question. In this study, we did not consider the precipitation, temperature, and increasing CO₂ concentration separately and then to evaluate the contribution of each factor. We considered climate change and increasing CO₂ concentration together as an integral impact factor. In current study, we focused on the issues of grazing intensity threshold detection for grassland degradation on the QTP under the changes of the external environment (including climate and CO₂ concentration together) as a whole. We agree with the reviewer that the contribution of different factors to grassland productivity and to the offset effects on the grassland of QTP is worth studying in depth

in the future.

We added a short discussion on this issue in the potential limitation as below in the revised version. (see lines 323-325).

“Evaluating the individual contribution of different factors (e.g. precipitation, temperature and raising CO₂ concentration) to the thresholds also could make a better understanding in grassland ecosystem degradation mitigation under climate change conditions.”

In this study, we did not turn off all other plant functional types, and only analyzed the results related to the plant function type of grass to focused on the issues of grazing intensity threshold for grassland degradation on the QTP.

For the point that the uncertainty of CO₂ fertilization effects on C3 grasses (Reich et al., 2018) mentioned by the reviewer, we added this in the discussion section as a potential limitation as below in the revised version. (see lines 321-323).

“Refinement of our approach should also consider uncertainties in how grasses respond to increasing CO₂ concentration⁵⁵ and to potential constrains of nutrients (e.g. nutrient removal by grazing).”

Also, in the revised version, we added more information about the model description on the vegetation dynamical component and nitrogen constrain process (Liu et al. 2005) as mentioned above. (also see lines 332-347).

There are no future change simulations such as those detailed in Tab.S1.1 that include grazing, and changing climate, but have CO₂ fixed. Such simulations would have allowed assessment of the magnitude to which the CO₂ fertilization effect was responsible for the positive effects of climate change.

RE: For the future simulations, our aim was to investigate the patterns of grazing intensity threshold under different Representative Concentration Pathways (RCPs) and make a comparison with that under current climate and CO₂ concentration. The grazing intensity threshold was determined by multiple simulations with a grazing intensity gradient from 0 SU/ha/year to 15 SU/ha/year (increasing by a rate of 0.5 SU/ha/year). There were 124 simulations (4 scenarios * 31) all together at the scale of whole QTP. Different from the simulations listed in Table S1.1, we don't have annual grazing map for the period of 2020-2100 unfortunately, so we currently can't evaluate the offset effects of future climate change and CO₂ concentration.

In this study, we considered climate change and increasing CO₂ concentration together as an integral impact factor to see the changing patterns of grazing intensity threshold in the future. We agree that the climate change and CO₂ fertilization effects may have different contribution on the grazing intensity patterns changing, but currently, we only considered the changes in the external environment (including climate and CO₂ concentration) as a whole and focused on what kind of grazing intensity should be applied under that circumstance in grazing management.

We added a short discussion on this issue in the potential limitation as below in the

revised version. (see lines 323-325).

“Evaluating the individual contribution of different factors (e.g. precipitation, temperature and raising CO₂ concentration) to the thresholds also could make a better understanding in grassland ecosystem degradation mitigation under climate change conditions.”

Further, potential interactions between nutrients, plant growth and photosynthesis, and nutrient removal via grazing do not appear to be included in the model.

RE: For the issues mentioned by the reviewer that potential interactions between nutrients and plant growth, we currently only include the nitrogen constrain in the model. However, nutrient removal via grazing was not considered in the current version of the model, and we added this potential limitation in the discussion section as below in the revised version. (see lines 321-323).

“Refinement of our approach should also consider uncertainties in how grasses respond to increasing CO₂ concentration⁵⁵ and to potential constrains of nutrients (e.g. nutrient removal by grazing).”

Additionally, the grazing threshold used (NPP close to zero) is not defined well. I remain uncertain about what “close” means. This threshold would need to be accurately defined and extensively justified as it is a pivotal metric in the manuscript.

RE: Yes, in the revised version, we defined grazing intensity threshold for grassland degradation based on the indicator of NPP in the introduction section, with an accurate statement specified in the text as below. (see lines 107-112).

“We adopted net primary productivity (NPP) as an indicator of grassland degradation, which has proven to be a good estimator of ecosystem functioning³³ and land degradation³⁴. We assumed that grassland productivity would decrease with increasing GI, and we defined the GI threshold to be the GI at which NPP fell to 1% of the pregrazing level that was considered as an extreme degradation status for grassland (see Methods).”

Overall the authors attempt to answer a highly relevant question, but the model (perhaps) used is not yet developed to the extent to sufficiently answer it. In any case, the authors have not provided enough information for me to critically asses the methodology.

RE: We are sorry for missing the information on the model description. Yes, according to the reviewer’s comments, we added more information about the model in the methodology section for a better understanding of the modeling method as mentioned before (also see lines 332-347).

In the method section, we also added more information on how the threshold was detected as below in the revised version (see lines 480-502).

“We assumed that grassland productivity would decrease with increasing GI. In order to detect the GI threshold for grassland degradation in each grid cell, the model

was run under different GIs, from non-grazing ($GI = 0.0$) to maximum possible grazing ($GI = 15 \text{ SU ha}^{-1} \text{ year}^{-1}$), increasing in a step of $0.5 \text{ SU ha}^{-1} \text{ year}^{-1}$ for a period of 120 years from the grazing activities applied on (Supplementary Fig. S3.5). Altogether 31 simulations were performed over the entire QTP. When the NPP of a cell reached 1% of the baseline NPP before the GI was applied, we defined that GI to be the GI threshold for the grid cell, above which GI would lead to degradation based on the indicator of NPP (Supplementary Fig. S3.5). This simulation strategy was performed on each grassland grid cell ($\sim 10\text{km} \times 10\text{km}$) of the QTP. Using this approach, and 3720 annual NPP maps (31 scenarios \times 120 years) of grassland on the QTP were used to generate a map of GI threshold and a map showing how many years before the grassland in each grid cell at the indicated GI (“time until degradation”). These simulations under different GIs were run with average climate and CO_2 conditions between 1980 and 2017.

Using a similar approach, we generated GI threshold maps under future climate change. The modeling was run using GIs ranging from non-grazing ($GI = 0.0$) to maximum possible grazing ($GI = 15 \text{ SU ha}^{-1} \text{ year}^{-1}$) by increasing the grazing level in a step of $0.5 \text{ SU ha}^{-1} \text{ year}^{-1}$ under the scenarios of RCP 2.6, 4.5 and 8.5, corresponding to low, medium, or high emissions in the 21st century ⁷¹ (Supplementary Table S3.3). A fourth scenario was considered in which climate conditions and CO_2 concentration were held constant at their 2020 levels. A total of 31 simulations across QTP grasslands were performed for each scenario, resulting in GI threshold maps. Model driving data on precipitation, temperature, relative humidity, and radiation of three RCPs were averaged from 24 General Circulation Models (GCMs) at a spatial resolution of 0.5° (Supplementary Table S3.4).”

And, we provided the source code with example input data in the manuscript submission system for the reviewers’ information.

Minor comments:

There was a general lack of transparency in the description of model and methods.

RE: Yes, according to the reviewer’s comments, we added more information about the model in the methodology section (see lines 332-347). In the method section, we also added more information on how the threshold was detected (see lines 480-502).

And, we provided the source code with example input data in the manuscript submission system for the reviewers’ reference.

Text size often changes within sentences and at many points in the manuscript e.g. lines 71-72.

RE: Yes, we reformatted the text all through the paper in the revised version.

There were many grammatical and language errors which could have been easily identified prior to submission. I won’t list all of these as there are tools available that

could be used to flag such errors.

RE: Yes, we have double checked the text and a language editing service was also applied for this revised version.

Line 61 - “leading to degradation” a concise definition of degradation is necessary.

RE: Yes, for lines from 58 to 62 in original version, we removed some redundant information and rephrased this part as below in the revised version (see lines 51-56).

“Earlier in history, animal farmers tended to maintain relatively small herds just large enough to support their nomadic lifestyle, with some extra animals to buffer the effects of harsh weather. The grazing of these small herds was sustainable because it did not lead to irreversible grassland degradation, defined as a decline in the quality of plants and soils, as well as changes in ecosystem composition, structure, and function 10, 11.”

As mentioned above, we also added definition of grassland degradation used in this study in the final part of the introduction section as mentioned before. (also see lines 107-112).

Line 63 – “In parallel, the productivity of alpine grasslands has fallen by 30% since late 1990s 8, 12 “. The studies cited here are in relation to the QTP and the authors here indicate that productivity has decreased since the 1990s. The results presented by the authors (S1.1 a) indicate that productivity (NPP) has increased over the majority of the QTP for this approximate time period based on their simulations. I may have missed something; how do the authors explain this apparent contradiction?

RE: Thanks to the reviewer for pointing out this. We have double checked the references and found that Dai et al. (2019) cited this number (i.e. 30%) from Dong et al. (2012). However, we did not find the specific percentage for productivity decreasing of QTP, while the study of Dong et al. (2012) focused on the growing season changes on QTP and showed that there was a significant increasing trend for the regional average growing season length during last 50 years, which may imply an increasing productivity. After we reviewed more literatures, the productivity of grassland on QTP were actually reported increasing (Gang et al., 2014; Liu et al., 2019; Wang et al., 2022). After reviewing and double checking the collected references, we deleted this statement in the revised version.

Lines 91-100 Numerous changes in text size.

RE: Yes, we reformatted the text in the revised version.

Lines 91-100 This paragraph sets the reasoning for the present study. The issue mentioned is that it is not clear whether local measurements can be extrapolated to

regional scales. The authors should provide more detail about how their modelling approach can do this better and be more explicit about which processes they expect to be able to represent better with their approach.

RE: Yes, we rephrased this part and highlight the advantages of process-based modelling on the aspects of spatial and temporal scale extrapolation over large region as below in the revised version (see lines 96-104).

“As a result, so-called “process-based ecosystem modeling”, which integrates information about biogeophysics, biogeochemistry, plant phenology, vegetation dynamics, as well as cycling of carbon, water and energy on the land surface, has shown promise for simulating interactions among vegetation, climate, and human activities^{30, 31}, making it well-suited to detect productivity changes in specific ecosystems³². Conducting long-term simulations at grid scale across the whole QTP using a process-based ecosystem model that consider the interaction of different processes among atmosphere, vegetation and soil could provide more insights into the spatial heterogeneity of ecosystem changes and could be extended into the past or the future to take climate change into account.”

We also emphasized it in the discussion section as below in the revised version (see lines 281-286).

“For ensuring sustainable grazing across the entire QTP, our threshold approach based on simulations at the grid scale may prove more sophisticated and more effective in the long run, especially after further refinement, than traditional approaches based on large-scale inventories or local grazing data. Our GI threshold grids can be extended backward in time to assess whether areas have historically been overgrazed, and they can be projected into the future to estimate grazing sustainability in the face of changing climate and elevated CO₂.”

Lines 101-106 Numerous changes in text size.

RE: Yes, we reformatted the text in the revised version.

Lines 101-106 Given that one of the main objectives of the study was to define a grazing intensity threshold, it would be good to have this threshold defined and justified here. It remains uncertain to me exactly how this threshold is defined and what its justification is. Future change scenarios: “Spatial pattern of future grazing intensity thresholds on QTP grasslands” (line 186) & Fig. 6 (line 646) – to what extent are the simulated increases in grazing intensity thresholds driven by increasing CO₂ alone?

RE: Yes, according to reviewers’ comments, we added the definition of “threshold” and criterion of “grassland degradation” in the introduction section as below in the revised version (see lines 107-112).

“We adopted net primary productivity (NPP) as an indicator of grassland degradation, which has proven to be a good estimator of ecosystem functioning³³ and land degradation³⁴. We assumed that grassland productivity would decrease with increasing GI, and we defined the GI threshold to be the GI at which NPP fell to 1% of

the pregrazing level that was considered as an extreme degradation status for grassland (see Methods).”

In the method section, we added more information on how the threshold was detected as mentioned before (also see lines 480-502).

For the future simulations, our aim was to investigate the patterns of grazing intensity threshold under different Representative Concentration Pathways (RCPs) and make a comparison with that under current climate and CO₂ concentration. In this study, we just considered climate change and increasing CO₂ concentration together as an integral impact factor and did not separately evaluate the individual contribution of each factor (precipitation, temperature, and increasing CO₂) to the thresholds.

We totally agree that evaluating the individual contribution of different factors on grasslands productivity and subsequently the threshold changing at the whole QTP scale is an important issue for our further study. Currently, we focused on the grazing intensity threshold detection for grassland degradation on the QTP and considered the changes in the external environment (including climate and CO₂ concentration together) as a whole and emphasized what kind of grazing intensity should be applied under that circumstance in grazing management.

We added a short discussion on this issue in the potential limitation as below in the revised version. (see lines 323-325).

“Evaluating the individual contribution of different factors (e.g. precipitation, temperature and raising CO₂ concentration) to the thresholds also could make a better understanding in grassland ecosystem degradation mitigation under climate change conditions.”

Figure 5 is very difficult to understand.

RE: Yes, we revised Figure 5 and rephrased the corresponding section. We cited the percentage numbers as showed exactly in the figure into the main text to avoid confusion as below in the revised version. (see lines 171-196).

We made an additional simulation that the grazing intensity was changed to make ratio of GI to threshold was reduced to 0.6 in areas where the ratio was > 0.6. The results were also added to Figure 5 to show a suitable ratio of GI to threshold (60%, ranged from 50% to 70%) may maintain the balance between grassland protection and human demands. (also see lines 171-196).

“Simulations suggested that climate change and elevated atmospheric CO₂ concentration would partly offset the negative effects of grazing activities on 68.3% of QTP grasslands (Figure 5a, b and Supplementary Information 1), nearly all of which (98.5%) was subject to normal grazing and which accounted for 83.6% of the total grassland area subject to normal grazing (Figure 5b). Across these “offset” areas, 69.2% showed a ratio of GI to the threshold below 0.6, while 83.0% showed a ratio below 0.7 (Figure 5c). The beneficial offsets were observed in only 5.3% of overgrazed areas (Figure 5b), 76.9% of which featured ratios of GI to the threshold below 3.0 (Figure

5d). Among the remaining 31.7% of QTP grassland areas where climate change and elevated CO₂ were not predicted to offset negative effects of grazing (Figure 5b), 58.5% were overgrazed (Figure 5b), with ratios of GI to threshold from 2.0 to 6.0 (Figure 5e), and 41.5% were subject to normal grazing (Figure 5b), of which 71.3% showed a ratio of GI to threshold larger than 0.6 and 88.4% showed a ratio of GI to threshold larger than 0.5 (Figure 5f). Although the areas in “non-offset areas” under normal grazing, the positive effects of climate change and elevated CO₂ could not compensate the negative effects of grazing on grassland productivity.

Our model suggested that setting the ratio of GI to threshold between 0.5 and 0.7 would preserve 70-80% of the areas that were currently subject to normal grazing and were predicted to experience positive offsets from climate change and CO₂ (Figure 5c, g). This range from 0.5 to 0.7 showed the greatest potential for maximizing the areas subject to normal grazing or overgrazing (both currently in “non-offset areas”) that could benefit from positive offsets due to climate change or CO₂ (Figure 5b, e, f). When we reduced the ratio of GI to threshold to 0.6 in areas where the ratio was > 0.6 in an additional simulation, we found that the positive effects of climate change and elevated CO₂ could offset the negative effects of grazing on over 89.1% of QTP grassland areas (Figure 5h), by “converting” 70% of currently “non-offset areas” into “offset areas” (Figure 5b, h). The “converted” areas come from 78% of overgrazing area and 58% of normal grazing area (Figure 5b, h).”

The revised Figure 5 showed as below (Figure R3.1):

Figure R3.1 (also Figure 5) Ratios of actual GI to the threshold on QTP grasslands. a. Distribution of areas where the negative effects of grazing are predicted to be offset by the positive effects of climate change and elevated CO₂ on grassland NPP (“offset areas”) and areas where such offset is not predicted

(“non-offset areas”); b. The bar charts indicated the area percentage of “offset areas” and “non-offset areas” of the grassland on QTP, along with the area percentage (white numbers) of overgrazing (OG, red color) and normal grazing (NG, green color) over each region, as well as the area percentage of OG (red color; blue numbers) and NG (green color; yellow numbers) distributed in offset areas and non-offset areas respectively; c-d. The area percentage histogram and area percentage cumulative curve of the ratio of GI to GI threshold for the “offset areas” subject to normal grazing (NG, green color) (c) or overgrazing (OG, red color) (d); e-f. The area percentage histogram and area percentage cumulative curve of the ratio of GI to GI threshold for the “non-offset areas” of OG (red color) (e) or NG (green color) (f), and different from others, the area percentage of GI ratio in panel f accumulated decreasingly from 0.95 to 0.15; g. The merged area percentage cumulative curves of the ratio of GI to GI threshold subject to NG (green color), extracted from panel c (blue line) and f (red line). h. The bar charts indicate the area percentage of “offset areas” and “non-offset areas” of QTP grasslands, along with the area percentage (white numbers) of OG (red color) or NG (green color) over each region based on an additional simulation that was performed in which the ratio of GI to threshold was set to be 0.6 where the ratio was > 0.6 .

Lines 434 – 445 It is difficult to understand. It is not clear to me what “When this comparison was repeated at the regional level after clustering sites by location” means and what was done. It is apparent from Fig. S2.6 Tab. S2.2 that the simulated grazing intensity threshold is almost always above the grazing level at the site, ca. 25/38. The authors write that potential causes for this may be the inconsistent spatial scale between the modelling grid (10km * 10km) and the grazing experimental site. Does the model simulate a 10x10 km patch of vegetation, or are they simulating something smaller (e.g. 1 Ha) and assuming this is representative of a 10x10km grid? What was the spatial scale of the site-level grazing experiments? The authors also mention that the site-level grazing experiments may have a grazing intensity that is in excess of the intensity threshold. How do they justify this? There are presumably measurements collected at these sites, e.g. grass growth (NPP), that could be used to justify this assumption. Based on the threshold methodology of the authors, wouldn't exceeding this threshold imply grass NPP at these sites is zero? Have the authors considered the alternative possibility that their model is wrong?

RE: Thanks for the valuable points. We rephrased this section in the revised version. The comparison was made at a county scale (grouped the site with county location). We revised the Figure S2.6 (Figure S3.4 in the revised version) and marked the group setting (original Table S2.2, revised Table S3.2) on the revised Figure S3.4 (original Figure S2.4).

We also realized that the designed heavy level of grazing intensity in the field grazing experiment may not have substantial connection with our theoretical grazing intensity threshold. The grazing intensity designed in the different field experiment may have different criterion and could have large variability among different sites, even though these sites are close to each other or located at the same position (Table S3.2). We did the simulation at a 10km×10km spatial scale, and we can see different sites with large variation of designed grazing intensity may located in the same modeling grid cell (Table S3.2).

In the revised version, we stated that comparison between the heavy grazing levels designed in the field studies and grazing intensity thresholds simulated in this study as a reference to justify the modeled grazing intensity threshold, which is not a true meaning of model validation.

We stated in the original version of MS that the site-level grazing experiments may have a grazing intensity that is in excess of the intensity threshold. It is a possibility inference since there is no clear relation between GIs defined as “heavy” in the literature and GI thresholds defined in our study. So, we removed this statement in the revised version to avoid any misunderstanding.

The reviewer had a concern whether the grassland NPP would reach zero when the designed heavy grazing intensity exceeded the modeled grazing intensity threshold. In the revised version, we made a clear statement that when the grassland NPP reduces to 1% of the original NPP level before grazing applied with an application of a certain GI, then that GI was defined as the GI threshold. We also rephrased the statement of “close to zero” in the revised version to avoid misunderstanding.

In fact, grassland degradation could be divided into different catalogs, including lightly degraded, moderate degraded and heavily degraded grasslands. In this study, we focus on heavily degraded scenarios, the GI threshold predicted to lead to an extreme degradation, which would cause ecosystem collapse. In our study, we also evaluated the time until degradation when we proposed the grazing intensity threshold, which implied that the grassland NPP would not decrease to the specified low level immediately while grazing was applied at the intensity of threshold.

Based on the comments of the reviewer, as well as the analysis mentioned above, we rephrased this section as below in the revised version (see lines 505-517).

“We make a comparison between the heavy grazing levels designed in the field experiment and GI thresholds simulated in this study as a reference to evaluate the modeled GI threshold (Supplementary Fig. S3.4). Modeled GIs were compared at the county level with “heavy” GIs at 38 field sites (Supplementary Fig. S3.4, Tables S3.2) in local grazing experiments in which sites were subjected to “light”, “medium” or “heavy” GI. Although this comparison may be imperfect because GIs defined as “heavy” in the literature have an unclear relation to GI thresholds defined in our study, regional mean simulated GI thresholds were quite consistent with designed heavy grazing intensities ($R^2 = 0.7853$, $p < 0.01$; Supplementary Fig. S3.6). Generally, the heavy GIs in experimental studies were slightly greater than the simulated GI thresholds (Supplementary Fig. S3.6). The difference in spatial scale between our modeling grid ($\sim 10 \times 10$ km) and local grazing experiment site (\sim several hectares) meant that the same modeling grid sometimes contained multiple sites differing substantially in GIs designated as heavy in experimental studies (Supplementary Table S3.2).”

Supplement

Fig. S2.6 - what is a) and what is b)? This is not specified.

RE: Yes, we revised the Figure S2.6 (Figure S3.4 in the revised version) and marked

the group setting (original Table S2.2, revised Table S3.2) on the revised Figure S3.4 (original Figure S2.4).

Fig. S2.7 – the difference between observed and simulated grass biomass seems rather large, up to 300% at some sites and grazing intensities. The “biomass for grazing” equations line 330/331 did not appear correctly. Perhaps this is just a rendering issue, though.

RE: The equations of biomass for grazing was adopted from the approach provided in the study of Shiyomi et al. (2011) and we revised the function only when above ground biomass felled in the range from $BioGrz_t \times 0.95$ to $BioGrz_t \times 0.95 \times 2.0$ in order to make the magnitude of biomass consumption in a continues way.

We also see there is a large difference between observed and simulated grass biomass at site of Maqu. Thank you for pointing out this. We double checked the five references in which the sites were all declared locating in Maqu county.

The original information of the five sites extracted from the references is:

ID	Longitude	Latitude	Group setting		Designed GI at heavy level(SU/ha/year)	Simulated GI threshold (SU/ha/year)	Data Source
			ID	County			
22	101.8833	35.9667	G4		3.20	5.50	Tan, 2012
23*	101.8830	35.9670	G4		7.70	5.50	Hu, 2015
24	101.8833	33.9667	G4	Maqu	16.00	9.00	Liu, 2018
25*	102.1172	33.7058	G4		9.60	9.50	Li et al., 2011
26*	101.7667	33.6667	G4		8.58	9.00	Zou et al., 2015

If we put them on the map (Figure R3.2), we can see point 22 and 23 are in the wrong place, since Maqu county (the yellow region) is far below these two points (Figure R3.2). Point 24 to 26 are in the right place of Maqu county. We are sure that the original coordinate information in the references (Tan, 2012; Hu, 2015) for point 22 and point 23 must be wrong. The position should be 101.883E, 33.967N, which is the same as point 24 (Figure R3.2). As a result, we extracted the wrong biomass data for point 23 in previous version, which induced the difference in the comparison between simulation and observation. In the revised version, we made a correction for the comparison at point 23 (Figure R3.3, also Figure S3.7). We can see the overall agreement between observed and simulated biomass was also improved (Figure R3.3f, also Figure S3.7f).

Figure R3.2 Sites location checking for Maqu county.

Figure R3.3 (also Figure S3.7) Comparison between observed aboveground biomass (red) at experimental field sites and the corresponding simulated aboveground biomass (blue) on grasslands subject to light, medium or heavy grazing intensity.

We updated Table S3.2 with the correct position information for points 22 and 23, as well as the GI threshold for these two points as below (also see Table S3.2).

ID	Longitude	Latitude	Group setting		Designed GI at heavy level(SU ha ⁻¹ year ⁻¹)	Simulated GI threshold (SU ha ⁻¹ year ⁻¹)	Data Source
			ID	County			
22	101.8833	33.9667	G4	Maqu	3.20	9.00	Tan, 2012
23*	101.8830	33.9670	G4		7.70	9.00	Hu, 2015
24	101.8833	33.9667	G4		16.00	9.00	Liu, 2018
25*	102.1172	33.7058	G4		9.60	9.50	Li et al., 2011
26*	101.7667	33.6667	G4		8.58	9.00	Zou et al., 2015

We also updated the map of sites distribution (Figure S3.4)

Grazing intensity threshold with NPP close to zero – based on the NPP equation (line 303), wouldn't zero NPP imply GPP was equal to maintenance respiration and there is no growth? Detailed justification for the choice of threshold is necessary as this is

crucial to the manuscript. At what time-step is grass growth calculated? At what time-step grass biomass removed by grazing? I can't figure out whether this happens at daily, monthly, or yearly time steps.

RE: From the view of the equation, zero NPP could be happened under the condition that GPP was equal to maintenance respiration. In our study, we detected the grazing threshold for grassland degradation using the indicator of NPP (specified to 1% of the original NPP level before grazing applied). We can see the NPP will gradually decrease while grazing was applied at intensity of threshold, and finally reached a low level (Figure S3.5). We considered this “near zero” (we rephrased the statement of “close to zero” in the revised version to avoid misunderstanding) level of NPP as an indicator for grassland extreme degradation while comparing to its original level before grazing activity was applied.

According to the reviewer's comments, in the introduction section, we defined grazing intensity threshold for grassland degradation based on the indicator of NPP as below in the revised version (see lines 107-112).

“We adopted net primary productivity (NPP) as an indicator of grassland degradation, which has proven to be a good estimator of ecosystem functioning³³ and land degradation³⁴. We assumed that grassland productivity would decrease with increasing GI, and we defined the GI threshold to be the GI at which NPP fell to 1% of the pregrazing level that was considered as an extreme degradation status for grassland (see Methods).”

And, the information on time-step (a daily time step) of grass growth and biomass removed by grazing was also added in the method section as below (see lines 477-479).

“Grassland productivity, biomass removed by grazing, and livestock body weight were updated using a daily time step in the model.”

Related References:

- Bardgett, R. D., Bullock, J. M., Lavorel, S., Manning, P., Schaffner, U., Ostle, N., . . . Shi, H. X. (2021). Combatting global grassland degradation. *Nature Reviews Earth & Environment*, 2(10), 720-735. doi:10.1038/s43017-021-00207-2
- Cai, H. Y., Yang, X. H., & Xu, X. L. (2015). Human-induced grassland degradation/restoration in the central Tibetan Plateau: The effects of ecological protection and restoration projects. *Ecological Engineering*, 83, 112-119. doi:10.1016/j.ecoleng.2015.06.031
- Cao, J. J., Adamowski, J. F., Deo, R. C., Xu, X. Y., Gong, Y. F., & Feng, Q. (2019). Grassland Degradation on the Qinghai-Tibetan Plateau: Reevaluation of Causative Factors. *Rangeland Ecology & Management*, 72(6), 988-995. doi:10.1016/j.rama.2019.06.001
- Dai, L., Guo, X., Ke, X., Zhang, F., Li, Y., Peng, C., . . . Du, Y. (2019). Moderate grazing promotes the root biomass in Kobresia meadow on the northern Qinghai-Tibet Plateau. *Ecology and Evolution*, 9(16), 9395-9406. doi:10.1002/ece3.5494
- Dong, M. Y., Jiang, Y., Zheng, C. T., & Zhang, D. Y. (2012). Trends in the thermal growing season throughout the Tibetan Plateau during 1960-2009. *Agricultural and Forest Meteorology*, 166, 201-206. doi:10.1016/j.agrformet.2012.07.013
- Dong, Q. M., Zhao, X. Q., Wu, G. L., Shi, J. J., & Ren, G. H. (2013). A review of formation mechanism and restoration measures of "black-soil-type" degraded grassland in the Qinghai-Tibetan Plateau. *Environmental Earth Sciences*, 70(5), 2359-2370. doi:10.1007/s12665-013-2338-7
- Feng, R. Z., Long, R. J., Shang, Z. H., Ma, Y. S., Dong, S. K., & Wang, Y. L. (2010). Establishment of *Elymus natans* improves soil quality of a heavily degraded alpine meadow in Qinghai-Tibetan Plateau, China. *Plant and Soil*, 327(1-2), 403-411. doi:10.1007/s11104-009-0065-3
- Foley, J. A., Prentice, I. C., Ramankutty, N., Levis, S., Pollard, D., Sitch, S., & Haxeltine, A. (1996). An integrated biosphere model of land surface processes, terrestrial carbon balance, and vegetation dynamics. *Global Biogeochemical Cycles*, 10(4), 603-628.
- Gaitán, J. J., Bran, D. E., Oliva, G. E., Aguiar, M. R., Buono, G. G., Ferrante, D., . . . Maestre, F. T. (2018). Aridity and Overgrazing Have Convergent Effects on Ecosystem Structure and Functioning in Patagonian Rangelands. *Land Degradation & Development*, 29(2), 210-218. doi:10.1002/ldr.2694
- Gang, C. C., Zhou, W., Chen, Y. Z., Wang, Z. Q., Sun, Z. G., Li, J. L., . . . Odeh, I. (2014). Quantitative assessment of the contributions of climate change and human activities on global grassland degradation. *Environmental Earth Sciences*, 72(11), 4273-4282. doi:10.1007/s12665-014-3322-6
- Gibbs, H. K., & Salmon, J. M. (2015). Mapping the world's degraded lands. *Applied Geography*, 57, 12-21. doi:10.1016/j.apgeog.2014.11.024
- Hartvigsen, G. (2022). Carrying Capacity, Concept of. In *Reference Module in Life Sciences*: Elsevier.
- Kucharik, C. J., Foley, J. A., Delire, C., Fisher, V. A., Coe, M. T., Lenters, J. D., . . . Gower, S. T. (2000). Testing the performance of a Dynamic Global Ecosystem Model: Water balance, carbon balance, and vegetation structure. *Global Biogeochemical Cycles*, 14(3), 795-825.
- Kwon, H.-Y., Nkonya, E., Johnson, T., Graw, V., Kato, E., & Kihui, E. (2016). Global Estimates of the Impacts of Grassland Degradation on Livestock Productivity from 2001 to 2011. In E.

- Nkonya, A. Mirzabaev, & J. von Braun (Eds.), *Economics of Land Degradation and Improvement – A Global Assessment for Sustainable Development* (pp. 197-214). Cham: Springer International Publishing.
- Liu, J., Price, D., & Chen, J. (2005). Nitrogen controls on ecosystem carbon sequestration: a model implementation and application to Saskatchewan, Canada. *Ecological Modelling*, *186*(2), 178-195. doi:10.1016/j.ecolmodel.2005.01.036
- Liu, J., Xu, X., & Shao, Q. (2008). The Spatial and Temporal Characteristics of Grassland Degradation in the Three-River Headwaters Region in Qinghai Province (in Chinese with English abstract). *Acta Geographica Sinica*, *63*(4), 364-376.
- Liu, Y. Y., Wang, Q., Zhang, Z. Y., Tong, L. J., Wang, Z. Q., & Li, J. L. (2019). Grassland dynamics in responses to climate variation and human activities in China from 2000 to 2013. *Science of the Total Environment*, *690*, 27-39. doi:10.1016/j.scitotenv.2019.06.503
- Ma, Y., & Lang, B. (1999). Review and Prospect of the Study on Black Soil Type Deteriorated Grassland (in Chinese with English abstract). *Pratacultural Science*, *16*(2), 5-9.
- McDonald, M. (2021). Ecological Security: A Definition. In M. McDonald (Ed.), *Ecological Security: Climate Change and the Construction of Security* (pp. 94-122). Cambridge: Cambridge University Press.
- Oñatibia, G. R., Amengual, G., Boyero, L., & Aguiar, M. R. (2020). Aridity exacerbates grazing-induced rangeland degradation: A population approach for dominant grasses. *Journal of Applied Ecology*, *57*(10), 1999-2009. doi:10.1111/1365-2664.13704
- Reich, P. B., Hobbie, S. E., Lee, T. D., & Pastore, M. A. (2018). Unexpected reversal of C3 versus C4 grass response to elevated CO2 during a 20-year field experiment. *Science*, *360*(6386), 317-320. doi:doi:10.1126/science.aas9313
- Ren, G. H., Shang, Z. H., Long, R. J., Hou, Y., & Deng, B. (2013). The relationship of vegetation and soil differentiation during the formation of black-soil-type degraded meadows in the headwater of the Qinghai-Tibetan Plateau, China. *Environmental Earth Sciences*, *69*(1), 235-245. doi:10.1007/s12665-012-1951-1
- Sheehy, D. (2001). The rangelands, land degradation and black beach: a review of research reports and discussions. In N. van Wageningen & W. Sa (Eds.), *The Living Plateau: Changing Lives of Herders in Qinghai* (pp. 5-9). ICIMOD: Kathmandu.
- Shiyomi, M., Akiyama, T., Wang, S., Yiruhan, Ailikun, Hori, Y., . . . Yamamura, Y. (2011). A grassland ecosystem model of the Xilingol steppe, Inner Mongolia, China. *Ecological Modelling*, *222*(13), 2073-2083. doi:10.1016/j.ecolmodel.2011.03.028
- van der Schrier, G., Barichivich, J., Briffa, K. R., & Jones, P. D. (2013). A scPDSI-based global data set of dry and wet spells for 1901-2009. *Journal of Geophysical Research-Atmospheres*, *118*(10), 4025-4048. doi:10.1002/jgrd.50355
- Wang, X. X., Dong, S. K., Sherman, R., Liu, Q. R., Liu, S. L., Li, Y. Y., & Wu, Y. (2015). A comparison of biodiversity-ecosystem function relationships in alpine grasslands across a degradation gradient on the Qinghai-Tibetan Plateau. *Rangeland Journal*, *37*(1), 45-55. doi:10.1071/rj14081
- Wang, Y., Lv, W., Xue, K., Wang, S., Zhang, L., Hu, R., . . . Niu, H. (2022). Grassland changes and adaptive management on the Qinghai-Tibetan Plateau. *Nature Reviews Earth & Environment*. doi:10.1038/s43017-022-00330-8
- Wu, G. L., Du, G. Z., Liu, Z. H., & Thirgood, S. (2009). Effect of fencing and grazing on a Kobresia-

- dominated meadow in the Qinghai-Tibetan Plateau. *Plant and Soil*, 319(1-2), 115-126. doi:10.1007/s11104-008-9854-3
- Wu, G. L., Ren, G. H., Dong, Q. M., Shi, J. J., & Wang, Y. L. (2014). Above- and Belowground Response along Degradation Gradient in an Alpine Grassland of the Qinghai- Tibetan Plateau. *Clean-Soil Air Water*, 42(3), 319-323. doi:10.1002/clen.201200084
- Zhang, K., Peng, C., Wang, M., Zhou, X., Li, M., Wang, K., . . . Zhu, Q. (2017). Process-based TRIPLEX-GHG model for simulating N₂O emissions from global forests and grasslands: Model development and evaluation. *Journal of Advances in Modeling Earth Systems*, 9(5), 2079-2102. doi:10.1002/2017ms000934
- Zhou, H., Yang, X., Zhou, C., Shao, X., Shi, Z., Li, H., . . . Ma, L. (2023). Alpine Grassland Degradation and Its Restoration in the Qinghai–Tibet Plateau. *Grasses*, 2(1), 31-46.
- Zhu, Q., Liu, J., Peng, C., Chen, H., Fang, X., Jiang, H., . . . Zhou, X. (2014). Modelling methane emissions from natural wetlands by development and application of the TRIPLEX-GHG model. *Geoscientific Model Development*, 7(3), 981-999. doi:10.5194/gmd-7-981-2014

Reviewer comments, second round:

Reviewer #2 (Remarks to the Author):

The manuscript now entitled: A productivity-based grazing intensity threshold as the early warning signal for grassland degradation on the Qinghai-Tibetan Plateau, has substantially improved from its previous version. The authors have responded to previous reviews satisfactorily and have addressed my main concern. The manuscript is now much clearer and maintains an adequate length.

A minor comment is that the authors in one of their responses confuse aridity with drought trend, and argue in relation to the trend when the suggestion pointed to how the spatial variation in aridity interacts with grazing intensity. I suggest incorporating into the discussion section how grazing intensity and aridity can interact.

Reviewer #3 (Remarks to the Author):

Dear Authors,

Thank you for your responses, model dynamics clarification, and code provision, and congratulations on a much-improved manuscript.

My previous main point of contention was that the CO₂ fertilisation of plant growth might strongly influence the model results. I would be more convinced of the solidity of your results had you chosen to run fixed CO₂ simulations, but I concede that these may be beyond the current scope of the manuscript. I hope you do run them.

The clarification on model dynamics in your response letter and in the manuscript has increased my confidence that CO₂ fertilisation effects are considered/constrained within your model; these constraints will therefore reduce the extent to which CO₂ is driving your modelled responses and reduce the need for fixed CO₂ simulations to examine the influence of CO₂ effects to some extent. The paper by LIU et al. (2022) (DOI: 10.5814/j.issn.1674-764x.2022.01.001) provided a very helpful overview of the IBIS model family. There appears to be author overlap. Consider providing this as an additional reference to describe your model.

I did not get your code running. The code appears complete and executable with adequate instructions. The difficulty was on my side as the necessary Fortran-netcdf libraries were not installed on my standard clusters, and MAC has changed usr/lib paths on my personal computer.

In my original review, I stated your key results and their significance. Now that I understand your methodology/model better, I can see no major flaws in the study. In the future, improvements could be made to the vegetation dynamic component, grazer component, and the interaction between the two, but this does not detract from your manuscript's current novelty and significance.

Reviewer #4 (Remarks to the Author):

First, I concur with the original reviewer that the use of the term "Grazing Intensity threshold" is inappropriate. What the authors are actually talking about is stocking rate (SU's/ha/year). Their model appears to have identified stocking rates that would lead to degradation (i.e. 99% decline in NPP), and mapped how that stocking rate threshold varies across the plateau. This is different from Grazing Intensity, because grazing intensity is typically used to refer to the proportion of NPP grazed in a given year. It is unclear how the model is handling year to year stochasticity in rainfall and NPP, but typically these semiarid grasslands have a high degree of inter-annual variability in

NPP, which leads to variability in grazing intensity (high in dry years, wet in low years) if stocking rate remains constant. I would recommend that the authors not use the term grazing intensity, and instead refer to a "stocking rate threshold".

A second major concern I have is that it remains unclear to what extent the model predictions have been validated with field data. The authors state that the model assumes NPP declines with increasing grazing intensity, but exactly how this is mathematically implemented is not explained. In many grasslands, we see that as grazing intensity increases up to some optimal level, grazing actually increases NPP (i.e. the grazing optimization hypothesis of McNaughton 1984), but past that point, NPP begins to decline again. Does their model predict a grazing optimization level, and where is this relative to what they are calling the "grazing intensity threshold"? How has the model been calibrated in terms of the above to belowground partitioning of photosynthate, relative to traits of the dominant graminoids on the plateau? Does the distribution of grass traits and species vary spatially across the region, and does this affect grazing tolerance? It does not appear that the model accounting for this issue. Also, do the model predictions depend on the temporal pattern of rainfall over time, i.e. the timing and frequency of droughts? Or are model predictions just based on applying mean annual rainfall each year of the simulations? Overall, the authors need to better document how the model has been parameterized and validated for this ecosystem, in order to be confident that the 'stocking rate threshold' maps they have generated are actually reliable. To the extent they can do this, the study provides an extremely valuable analysis of how the current distribution of stocking rates in the region relates to likelihood of long-term degradation.

RESPONSES TO REVIEWERS' COMMENTS

Reviewer #2 (Remarks to the Author):

The manuscript now entitled: A productivity-based grazing intensity threshold as the early warning signal for grassland degradation on the Qinghai-Tibetan Plateau, has substantially improved from its previous version. The authors have responded to previous reviews satisfactorily and have addressed my main concern. The manuscript is now much clearer and maintains an adequate length.

A minor comment is that the authors in one of their responses confuse aridity with drought trend, and argue in relation to the trend when the suggestion pointed to how the spatial variation in aridity interacts with grazing intensity. I suggest incorporating into the discussion section how grazing intensity and aridity can interact.

RE: Thanks for your positive feedbacks. Yes, we added discussion about the interaction between grazing intensity and aridity in the revised version as below. (See lines 236-240).

“The most arid areas on northwest OTP were generally overgrazed and with low grazing intensity threshold, which partially reflected the interaction between the spatial variation of aridity and grazing pressure. Some studies pointed out that increasing aridity could exacerbate the negative effects of overgrazing on grassland ecosystem (Gaitán et al., 2018), and accelerate grassland degradation (Oñatibia et al., 2020).”

Added References:

- Gaitán JJ, *et al.* Aridity and Overgrazing Have Convergent Effects on Ecosystem Structure and Functioning in Patagonian Rangelands. *Land Degradation & Development* **29**, 210-218 (2018).
- Oñatibia GR, Amengual G, Boyero L, Aguiar MR. Aridity exacerbates grazing-induced rangeland degradation: A population approach for dominant grasses. *Journal of Applied Ecology* **57**, 1999-2009 (2020).

Reviewer #3 (Remarks to the Author):

Dear Authors,

Thank you for your responses, model dynamics clarification, and code provision, and congratulations on a much-improved manuscript.

RE: Thanks for your positive feedbacks.

My previous main point of contention was that the CO₂ fertilization of plant growth might strongly influence the model results. I would be more convinced of the solidity of your results had you chosen to run fixed CO₂ simulations, but I concede that these may be beyond the current scope of the manuscript. I hope you do run them.

RE: Thanks for the comments. We are sorry that we did not fully understand your comments in the previous version and did not state clearly in the method section. Actually, there are two types of simulations in our study.

Firstly, as showed in supplementary information section 1, in order to evaluate the effects of climate change, elevated CO₂ concentration and stocking rate on grassland NPP of QTP, we conducted the scenarios with or without varied meteorological data (daily), ramping CO₂ concentration data (yearly) and dynamic stocking rate data (yearly) from 1980 to 2017. Then we could evaluate the effects of grazing and climate change/elevated CO₂ concentration on grassland NPP of QTP by the differences between scenarios.

Secondly, in order to detect the stocking rate threshold on grassland of QTP (as showed in method section and Figure S3.5), we conducted an ensemble of simulations forced by a multiyear (1980-2017) mean daily meteorological and CO₂ concentration data during the modeling period of 120 years (Figure S3.5). That is to say, in the simulations of threshold detection, the climate and CO₂ condition was actually fixed, which avoided the influences of climate change and elevated CO₂ concentration on grassland growth. It was similar for the simulations in future period by using multiyear (2020-2100) average climate and CO₂ concentration data for each scenario.

In the revised methods section, we made a separated section for “stocking rate threshold detection” and clearly stated that we actually used fixed climate and CO₂ concentration data in the ensemble simulations for threshold detection. (See line 489, lines 500-503, and lines 516-517).

“During this period, multiyear (1980-2017) averages of daily meteorological and CO₂ concentration data were used to drive the model in order to eliminate the effects of varied climate and ramping CO₂ concentration on NPP in the process of threshold detection.”

“Multiyear (2020-2100) averaged climate and CO₂ concentration data for each scenario was used to drive the model.”

The clarification on model dynamics in your response letter and in the manuscript has increased my confidence that CO₂ fertilization effects are considered/constrained within your model; these constraints will therefore reduce the extent to which CO₂ is driving your modelled responses and reduce the need for fixed CO₂ simulations to examine the influence of CO₂ effects to some extent. The paper by LIU et al. (2022)(DOI: 10.5814/j.issn.1674-764x.2022.01.001) provided a very helpful overview of the IBIS model family. There appears to be author overlap. Consider providing this as an additional reference to describe your model.

RE: Thanks for the comments and suggestions. We actually conducted simulations with fixed climate and CO₂ to examine stocking rate threshold. We now state it clearly in the revised method section as mentioned above. Yes, we added the reference of Liu et al. (2022) in the revised version.

I did not get your code running. The code appears complete and executable with adequate instructions. The difficulty was on my side as the necessary Fortran-netcdf libraries were not installed on my standard clusters, and MAC has changed usr/lib paths on my personal computer.

RE: Yes, it is necessary to install Fortran-netcdf libraries (version 4.0 or later) on the machine. And additional environmental setting could be different among different types of clusters. Based on our experiences, it is a relatively easy way to test the model by installing Cygwin under windows operating system. Just select the latest packages of netcdf, hdf, gfortan during the Cygwin installation.

In my original review, I stated your key results and their significance. Now that I understand your methodology/model better, I can see no major flaws in the study. In the future, improvements could be made to the vegetation dynamic component, grazer component, and the interaction between the two, but this does not detract from your manuscript's current novelty and significance.

RE: Thank you very much for your valuable comments and suggestions. In the future, we will try to improve the model by considering grazer component and vegetation dynamic component, as well as the interaction between them and make further applications.

Reviewer #4 (Remarks to the Author):

First, I concur with the original reviewer that the use of the term “Grazing Intensity threshold” is inappropriate. What the authors are actually talking about is stocking rate (SU’s/ha/year). Their model appears to have identified stocking rates that would lead to degradation (i.e. 99% decline in NPP), and mapped how that stocking rate threshold varies across the plateau. This is different from Grazing Intensity, because grazing intensity is typically used to refer to the proportion of NPP grazed in a given year. It is unclear how the model is handling year to year stochasticity in rainfall and NPP, but typically these semiarid grasslands have a high degree of inter-annual variability in NPP, which leads to variability in grazing intensity (high in dry years, wet in low years) if stocking rate remains constant. I would recommend that the authors not use the term grazing intensity, and instead refer to a “stocking rate threshold”.

RE: Yes, according to the reviewer’s comments, we used the term of “stocking rate threshold” instead of “grazing intensity threshold” in the revised version. Changes were made in main text and supplementary information for all through the text, tables and figures. In this study, daily meteorological data including rainfall was used to drive the model and then output the grassland NPP.

A second major concern I have is that it remains unclear to what extent the model predictions have been validated with field data. The authors state that the model assumes NPP declines with increasing grazing intensity, but exactly how this is mathematically implemented is not explained. In many grasslands, we see that as grazing intensity increases up to some optimal level, grazing actually increases NPP (i.e. the grazing optimization hypothesis of McNaughton 1984), but past that point, NPP begins to decline again. Does their model predict a grazing optimization level, and where is this relative to what they are calling the “grazing intensity threshold”? How has the model been calibrated in terms of the above to belowground partitioning of photosynthate, relative to traits of the dominant graminoids on the plateau? Does the distribution of grass traits and species vary spatially across the region, and does this affect grazing tolerance? It does not appear that the model accounting for this issue. Also, do the model predictions depend on the temporal pattern of rainfall over time, i.e. the timing and frequency of droughts? Or are model predictions just based on applying mean annual rainfall each year of the simulations? Overall, the authors need to better document how the model has been parameterized and validated for this ecosystem, in order to be confident that the ‘stocking rate threshold’ maps they have generated are actually reliable. To the extent they can do this, the study provides an extremely valuable analysis of how the current distribution of stocking rates in the region relates to likelihood of long-term degradation.

RE: Thanks for the comments. For the model validation, in this study, we evaluated the model performance by comparing the simulated results with the collected field observations (including the stocking rates (See Fig. S3.6), the grassland biomass (See

Fig. S3.7)), the eddy covariance observations (GPP) (See Fig. S3.8), and the remote sensing NPP productions (NPP retrieved from MODIS and AVHRR) (See Fig. S3.9). We also made a comparison between our results and existing literatures on grazing and degradation status of grassland on the plateau (See supplementary information section 2). The validation indicated that the model is reliable and then we used the model to detect the stocking rate threshold with simulations under different scenarios.

For the concerns on NPP, in the revised version, according to the reviewer's comments, we added more details on the calculation of NPP declining due to grazing activities in the method section as below. (See lines 490-496).

"The consumption of NPP (NPP_{Grz}) at each daily timestep was calculated as the minimum of biomass consumed from live grass ($BioGrz_l$) that directly depended on SR and the NPP could be allocated to leaf:

$$NPP_{Grz} = \min(NPP_D * \alpha_{leaf}, BioGrz_l)$$

where NPP_D was daily net primary productivity of grassland ($kg\ C\ m^{-2}$), α_{leaf} was the allocation fraction of total photosynthate to leaf (Supplementary Table S3.1)."

As the reviewer mentioned, we did not include specific process of the possibility of compensatory grass growth in response to light or moderate grazing. On one hand, it still remains controversial and is difficult to quantify the compensatory effects in the model currently; on the other hand, in this study, we focused on detecting a stocking rate threshold that would induce extreme grassland degradation. Although there could be a grazing optimization level for grassland NPP under light or moderate grazing condition, we thought the compensatory effects of grazing was not a dominant factor for the estimation of stocking rate threshold that indicating an extreme heavy grazing condition. We also totally agree with the reviewer that it would be meaningful for grazing management if we could predict an optimization grazing level by integrating grazing optimization hypothesis in the model in the future. We discussed this topic in the discussion section as below. (See lines 314-320).

"Meanwhile, the possibility of compensatory grass growth in response to light or moderate grazing was not included in the simulations, primarily because this possibility remains controversial. Some studies have indicated that it is rare, or that it depends on the type and intensity of environmental stress factors or nutrient availability. Future studies could take compensatory growth into account and explore whether and how it may affect the overall grazing capacity of alpine grasslands as they undergo warming and humidification in the future."

In this study, in fact, we did not include specific classification of grass species and the grazing tolerance in the model calibration and simulations. In our current dynamic vegetation model, we considered two plant function types of grass including C3 and C4

grasses. According to the reviewer’s comments, we added this in the limitation discussion in the revised version as below. (See lines 329-331)

“Since climate change and grazing activities would affect the structure of palatable and unpalatable grasses, as well as the grazing tolerance, future model improvements could include detailed grass traits and species information instead of only two general grass PFTs.”

We added a table to list the major parameters including the allocation fraction of total photosynthate of these two grass plant function types in supplementary information (See Supplementary Table S3.1). The parameters including the allocation fraction of total photosynthate to leaf (aboveground) and root (belowground) were original from Kucharik et al. (2000) and adjustment was adopted from the study of Yuan et al (2011), in which parameters were optimized for the vegetation distribution simulation in China. Currently, the parameters of the allocation fraction of total photosynthate to leaf and root were uniform across the plateau for each plant function type. In the revised version, we added the above information in the method section as below. (See lines 356-358).

“The parameters of the two grass PFTs in the original IBIS model were adjusted to fit the natural vegetation distribution in China by Yuan et al. (2011) (Supplementary Table S3.1).”

And an additional table in supplementary information as below.

“

Table S3.1 Major parameters of plant functional types (PFT) of grasses

Plant function type	m	b	$V_{cmax,15}$	$T_w(^{\circ}C)$	GDD_0	σ (m^2kg^{-1})	α_{leaf}	α_{root}	τ_{leaf} (years)	τ_{root} (years)
Warm (C4) grasses	4.0	0.04	4.0	>22.0	>100	20.0	0.45	0.55	1.25	1.00
Cool (C3) grasses	9.0	0.01	25.0		>100	20.0	0.45	0.55	1.00	1.00

Note: m, slope of stomatal conductance relationship (nondimensional); b, intercept of the stomatal conductance relationship ($mol\ H_2O\ m^{-2}\ s^{-1}$); $V_{cmax,15}$, maximum Rubisco capacity of the top leaf ($\mu mol\ CO_2\ m^{-2}\ s^{-1}$) at 15°C; T_w , temperature of the warmest month; GDD_0 , growing degree days calculated on a 0°C base; σ , the specific leaf area; α_{leaf} and α_{root} , the allocation fraction of total photosynthate to leaf and root; τ_{leaf} and τ_{root} , the residence time of carbon in leaf and root. The parameters of the two grass PFTs in the original IBIS model (Kucharik et al. 2000) were adjusted to fit the natural vegetation distribution in China (Yuan et al. 2011).

”

Our model was driven with daily meteorological data including rainfall. The daily-scale data potentially contains the information of the timing and frequency of precipitation. In the revised version, we clearly stated the forcing data used for different simulations scenarios. (See lines 479-482, lines 500-503, and lines 516-517).

“For the simulations to evaluate the effects of climate change, elevated CO_2

concentration and stocking rate on NPP of QTP grasslands (Supplementary Information 1), the model ran with daily meteorological data beginning from 1960 to 2017, while grazing activity was applied from 1980 to 2017.”

“During this period, multiyear (1980-2017) averages of daily meteorological and CO₂ concentration data were used to drive the model in order to eliminate the effects of varied climate and ramping CO₂ concentration on NPP in the process of threshold detection.”

“Multiyear (2020-2100) averaged climate and CO₂ concentration data for each scenario was used to drive the model.”

Added References:

Kucharik CJ, *et al.* Testing the performance of a Dynamic Global Ecosystem Model: Water balance, carbon balance, and vegetation structure. *Global Biogeochem Cy* **14**, 795-825 (2000).

Yuan Q, Zhao D, Wu S, Dai E. Validation of the Integrated Biosphere Simulator in simulating the potential natural vegetation map of China. *Ecological Research* **26**, 917-929 (2011).